# Robust ultraclean atomically thin membranes for atomic-resolution electron microscopy

Liming Zheng[1,12], Yanan Chen[2,11,12], Ning Li[3,4,12], Jincan Zhang[1,4], Nan Liu[2,5], Junjie Liu[6], Wenhui Dang[1], Bing Deng[1], Yanbin Li[7], Xiaoyin Gao[1], Congwei Tan[1,4], Zi Yang[2], Shipu Xu[1], Mingzhan Wang[1], Hao Yang[1,4], Luzhao Sun[1,4], Yi Cui[7], Xiaoding Wei[6,8], Peng Gao[3,9]*, Hong-Wei Wang[2,5,10]* & Hailin Peng[1,4]*

The fast development of high-resolution electron microscopy (EM) demands a background-noise-free substrate to support the specimens, where atomically thin graphene membranes can serve as an ideal candidate. Yet the preparation of robust and ultraclean graphene EM grids remains challenging. Here we present a polymer- and transfer-free direct-etching method for batch fabrication of robust ultraclean graphene grids through membrane tension modulation. Loading samples on such graphene grids enables the detection of single metal atoms and atomic-resolution imaging of the iron core of ferritin molecules at both room- and cryo-temperature. The same kind of hydrophilic graphene grid allows the formation of ultrathin vitrified ice layer embedded most protein particles at the graphene-water interface, which facilitates cryo-EM 3D reconstruction of archaea 20S proteasomes at a record high resolution of ~2.36 Å. Our results demonstrate the significant improvements in image quality using the graphene grids and expand the scope of EM imaging.

[1] Center for Nanochemistry, Beijing Science and Engineering Center for Nanocarbons, Beijing National Laboratory for Molecular Sciences, College of Chemistry and Molecular Engineering, Peking University, 100871 Beijing, China. [2] Ministry of Education Key Laboratory of Protein Sciences, Beijing Advanced Innovation Center for Structural Biology, School of Life Sciences, Tsinghua University, 100084 Beijing, China. [3] International Center for Quantum Materials, and Electron Microscopy Laboratory, School of Physics, Peking University, 100871 Beijing, China. [4] Academy for Advanced Interdisciplinary Studies, Peking University, 100871 Beijing, China. [5] Tsinghua-Peking Joint Center for Life Sciences, Tsinghua University, 100084 Beijing, China. [6] State Key Laboratory for Turbulence and Complex System, Department of Mechanics and Engineering Science, College of Engineering, Peking University, 100871 Beijing, China. [7] Department of Materials Science and Engineering, Stanford University, Stanford, CA 94305, USA. [8] Beijing Innovation Center for Engineering Science and Advanced Technology, Peking University, 100871 Beijing, China. [9] Collaborative Innovation Center of Quantum Matter, 100871 Beijing, China. [10] Beijing Frontier Research Center for Biological Structures, Tsinghua University, 100084 Beijing, China. [11] Present address: School of Materials Science and Engineering, Tianjin University, Tianjin 300072, China. [12] These authors contributed equally: Liming Zheng, Yanan Chen, Ning Li. *email: p-gao@pku.edu.cn; hongweiwang@tsinghua.edu.cn; hlpeng@pku.edu.cn

Technological breakthroughs in electron microscopy (EM) have enabled the imaging of individual atoms at the sub-angstrom resolution for inorganic materials[1] and near-atomic resolution for biological specimen supported by EM grids[2]. However, atomic-resolution cryo-EM is still challenging when biomolecules are embedded in unsupported vitrified ice layer at cryogenic temperature. During the cryo-EM specimen preparation, unsupported protein particles tend to adsorb at the air–water interfaces, which might cause preferential orientation and denaturation of proteins, thus limited the attainable resolution of the reconstructions[3–5]. To address this issue, conventional amorphous carbon films were often used to support the proteins. However, their non-negligible background noise and radiation-induced specimen motion hamper the further improvement of image resolution[6].

Graphene, an atomically thin carbon two-dimensional crystal with high conductivity and mechanical strength, is an ideal specimen support to substantially minimize background noise[7–9], reduce radiation-induced sample motion[10], and especially avoid air–water interface adsorption of proteins during cryo-EM[5,11,12]. An ideal graphene grid as specimen support should exhibit several crucial characteristics: robust enough to avoid breakage[13,14]; clean surface to provide high contrast[15,16]; and tunable hydrophilicity for efficient specimens loading[10]. Although large-area high-quality graphene films can be grown on a metal substrate by chemical vapor deposition (CVD) and allow for the manufacturing of EM grids[17,18], scalable fabrication of robust and ultraclean graphene EM grids is still a great challenge. Current preparation methods mainly rely on the transfer of graphene onto the perforated EM grids[14,15,19–21], which always result in the low efficiency and low coverage of suspended graphene membrane. Furthermore, polymer films have often been coated on the graphene surface as supports during previous fabrication techniques, so that polymer residues on the graphene membranes would inevitably degrade the intrinsic performances of graphene[18,22] and introduce extra background noises in EM[19,21]. It is therefore critical to develop a robust and scalable method to produce ultraclean and hydrophilic graphene grids for more general application in high-resolution EM imaging.

Herein, we exploit a polymer- and transfer-free method for scalable fabrication of robust and ultraclean graphene grids from large-area CVD-grown graphene films on metal foil. Graphene grids show tunable surface hydrophilicity, which enables efficient specimens loading for atomic-resolution EM at both room- and cryo-temperature. Such grids have also been proved to achieve ultrathin vitrified ice layer (~30 nm in thickness) with most protein particles surrounded at the graphene–water interface, achieving 3D reconstruction of archaea 20S proteasome at a record high resolution of 2.36 Å by single-particle cryo-EM[23–28].

## Results

**Design of robust suspended graphene membranes**. Figure 1a shows the scalable fabrication procedure of clean graphene EM grids. Briefly, large-area graphene film was first grown on the Cu foil by CVD method with a surface coverage of 100%. Then the backside of Cu foil (without graphene covering) was etched directly into arrayed Cu grids with photolithography (Supplementary Fig. 1). After rinsing and drying, the graphene membrane suspended on the spaced holes was formed with an ultraclean surface, thanks to the transfer-free and polymer-free process.

The robustness of suspended graphene membranes is essential for an EM specimen support. We analyzed interfacial forces on graphene membranes induced by the solution during the drying process (Fig. 1b). Surface tension may lead to mechanical failure at linear defects of suspended graphene (Supplementary Fig. 2), since the

linear defects such as wrinkles and low-angle tilt boundaries of graphene seriously degrade its mechanical properties[29–32]. The graphene membrane bent down due to the capillary pressure $P_{cap}$ induced by the liquid on the grid holes. $P_{cap}$ can be estimated according to Young-Laplace equation $P_{cap} = 2\gamma_{l-g}/R_l$, where $\gamma_{l-g}$ is the surface tension of the liquid and $R_l$ is the curvature radius of the liquid surface. Since the grid hole has an inclined wall from fabrication, $R_l$ decreases during the drying process, leading to an increasing $P_{cap}$ accordingly. Thus, the deformation of the graphene membrane will increase gradually (Supplementary Fig. 3). The maximum stress of the membrane induced by the surface tension was estimated to be ~10 GPa during the liquid volatilization in a 30–50-μm-sized grid hole (Supplementary Table 1, Supplementary Note 1, Supplementary Equations 1 and 2). If the membrane was not robust enough to withstand the surface tension (Fig. 1b, Supplementary Figs. 4 and 5, Supplementary Movie 1), it would have eventually broken around the linear defects, where the stress concentration occurred (Fig. 1c)[29–32]. Considering the theoretical strength of pristine graphene monolayer is ~120 GPa[33], high-quality monolayer graphene membrane without linear defects will remain intact during drying (Fig. 1b, c, Supplementary Fig. 6, Supplementary Movie 2). Moreover, according to the theory of elasticity, the radial stress $\sigma$ within the membrane can be expressed as $\sigma = P_{cap} a^2 / 4ht$, where $t$ is the thickness of the membrane[34]. For the bilayer or few-layer graphene whose thickness $t$ increases, the stress in the membrane reduces significantly, which helps to effectively avoid the rupture of graphene grids (Fig. 1b, Supplementary Fig. 7, Supplementary Movie 3). Thus, the tension of graphene films can be modulated by adjusting the layer number and quality to acquire high-intactness and robust graphene membranes (Fig. 1c).

Through the membrane tension modulation, we successfully fabricated graphene EM grids in an array as large as $18 \times 18$ grids in counts from a 64 cm$^2$ Cu foil with a thickness of ~25 μm (Fig. 1d). Each graphene grid typically contains about 800 highly ordered holes with controlled morphology and a lateral diameter of 20–50 μm depending on the photolithography pattern design (Fig. 1e, Supplementary Figs. 8 and 9). The grid can be handled as easy as regular EM grid, and the large-area suspended graphene membrane provides enough space to load samples, allowing efficient EM characterization. The bilayer and few-layer graphene grids can reach the intactness as high as 91–100% after being dried in the air (Fig. 1f, Supplementary Fig. 10). To further decrease the surface tension of the liquid and increase the intactness of monolayer membrane, the critical-point drying can be applied to improve the intactness up to ~90% for single-crystal monolayer and ~78% for polycrystalline monolayer graphene grids, respectively (Fig. 1f, Supplementary Fig. 11). Moreover, the smaller suspended size of graphene membranes also contributes to the lower probability of breakage (Supplementary Fig. 12).

**Ultraclean graphene membranes with low background noise**. Our transfer-free graphene grids exhibit ultraclean surfaces with negligible background noise, in contrast to the conventional transferred graphene grids assisted with polymer film coating (Supplementary Fig. 13). As shown in Fig. 2a, b, high-magnification high-angle annular dark-field scanning transmission electron microscopy (HAADF-STEM) images from representative areas on the graphene grids showed no polymer contamination, and the graphene lattice was distinctly presented by the hexagonally arranged carbon atoms. The Fast Fourier Transform (FFT) pattern of the raw image (inset in Fig. 2b) shows that the ultraclean graphene membrane can be imaged with the spatial resolution better than 1.07 Å, which is the best result of graphene imaging at 60 kV[35]. Owing to the ultraclean and ultrathin features of our graphene membranes, the

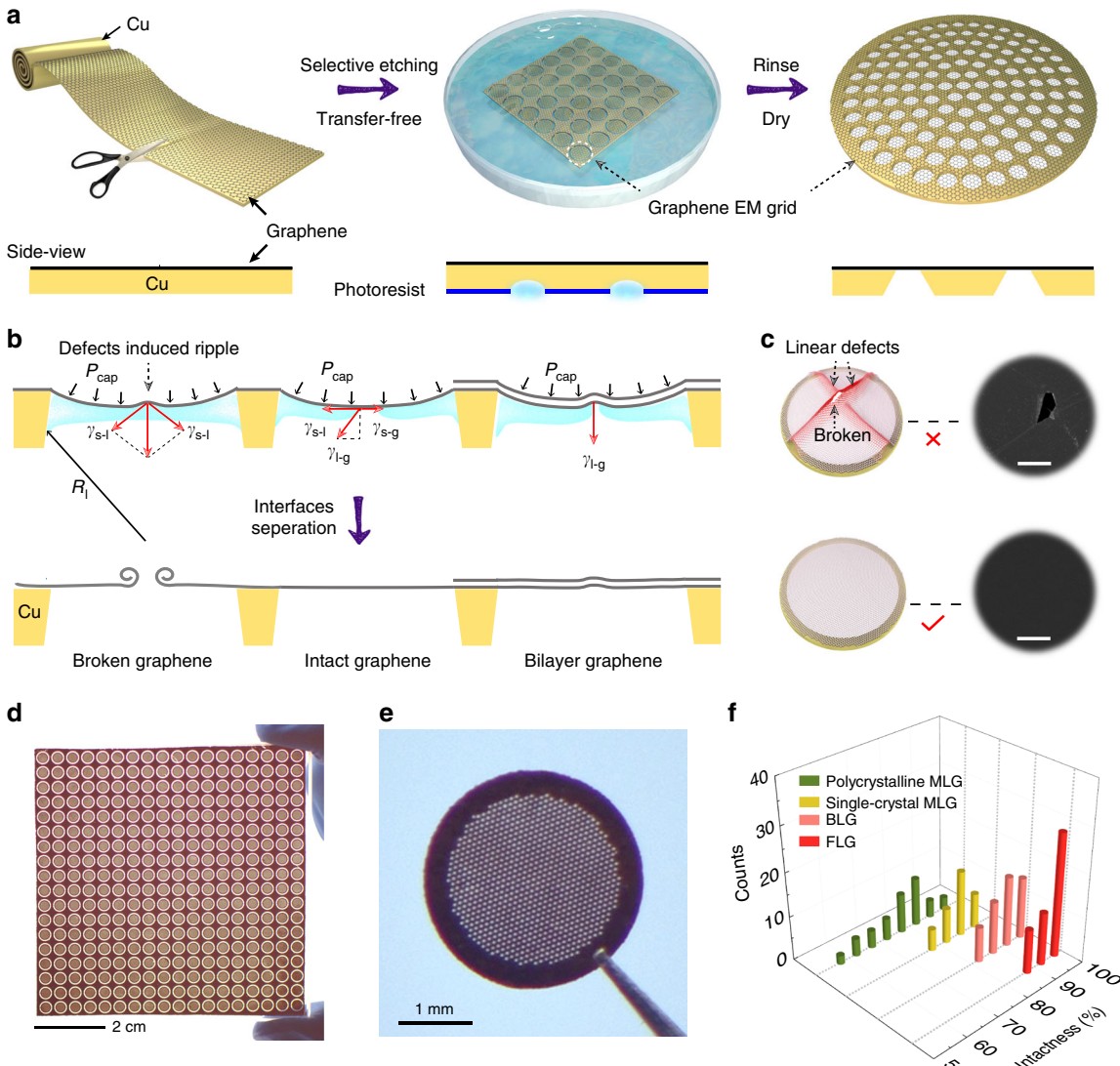

**Fig. 1 Scalable fabrication of robust and ultraclean graphene EM grids. a** Schematic representation illustrating the ultraclean graphene grids prepared in a batch and transfer-free way: large-area and continuous graphene film is grown on Cu foil, then the Cu backside with patterned photoresists is selectively etched; after rinsing and drying, graphene grids are prepared. **b** Interfacial forces on the graphene during the drying process. Under the capillary pressure and the surface tension of the liquid, the rupture of graphene occurs around the defect area, while the graphene without defects and the bilayer graphene tends to keep intact. **c** Schematic of graphene mechanical failure from the linear defects such as wrinkles and boundaries after drying (top row). To fabricate the intact suspended graphene membrane, graphene without linear defects is more feasible (bottom row). Scale bars, 4 μm. **d** Photograph of the resultant arrayed graphene grids (18 × 18 in counts on 64 cm² Cu foil). **e** Photograph of a typical graphene EM grid consisting of hexagonally arranged holes. **f** The intactness statistics of different graphene grids. The suspended graphene membranes are single-crystal monolayer graphene (MLG), polycrystalline MLG, bilayer graphene (BLG) and few-layer graphene (FLG), respectively. Source data are provided as a Source Data file.

background level of the electron energy loss spectroscopy (EELS) of the graphene membrane is much lower than that of amorphous carbon film (Fig. 2c). Such ultraclean graphene membrane enabled atomic-resolution imaging of fine particles loaded on it (Supplementary Fig. 14) due to the weak scattering from the atomically thin carbon support. Remarkably, individual Cu atoms and clusters loaded on graphene membranes can be clearly resolved (Fig. 2d, Supplementary Fig. 15) with high contrast and negligible background noise (Fig. 2e); and even a single Cu atom can be easily identified by EELS due to the low background of the spectra (Fig. 2f).

**Structure and surface property manipulation of graphene membranes.** Tunable hydrophilicity of graphene grids is the key

to the efficient load of both hydrophobic and hydrophilic particle specimens in solutions. To this end, we regulated water contact angles (WCAs) of the graphene grids from $90 \pm 2°$ to $50 \pm 3°$ under controllable oxygen-plasma treatments within 40 s (Fig. 3a). Meanwhile, the site-defect density of graphene and the average distance between defects ($L_D$) can be modulated via adjusting the condition of plasma treatments. As shown in Fig. 3b, the intensity ratio of D to G peak ($I_D/I_G$) of Raman spectroscopy for graphene membrane maintained suppressed (~0.05) within 10 s of plasma treatments, and then $I_D/I_G$ distinctly increased to ~2 with a corresponding $L_D$ of ~10 nm as the treatment prolonged (Supplementary Fig. 16). Atomic-resolution image of the treated graphene clearly revealed the existence of oxygen (Fig. 3c), where the brighter atoms along X–X′ were confirmed to be O atoms by the intensity profile (Fig. 3d).

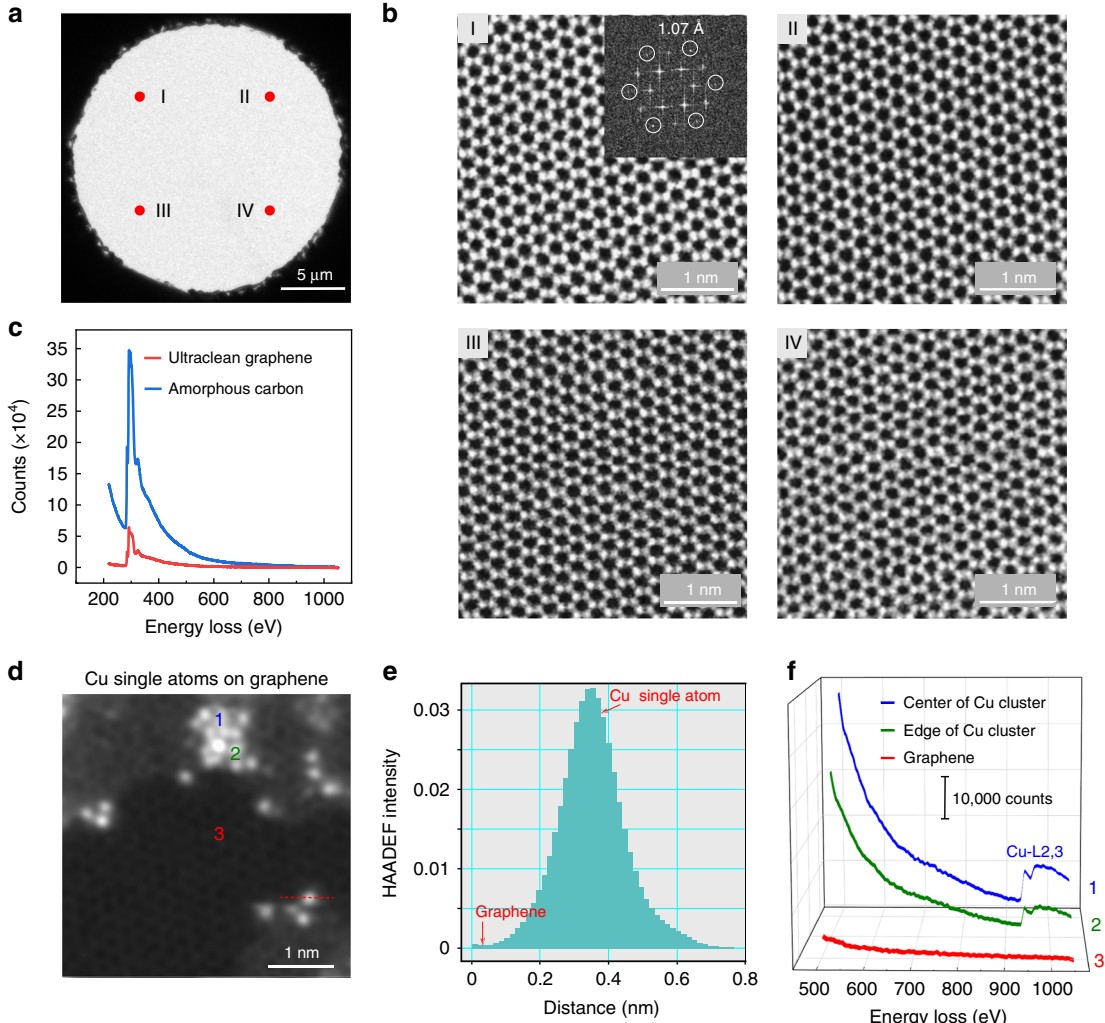

**Fig. 2 Ultraclean graphene membranes with low background noise. a** Low-magnification TEM image of typical suspended graphene membrane. **b** HAADF-STEM images from four marked regions (I, II, III, IV) shown in **a**. Inset of the panel I of **b**: Fast Fourier Transform (FFT) of the raw image, revealing a spatial resolution of 1.07 Å. **c** EELS spectra of the ultraclean graphene and amorphous carbon film. **d** HAADF-STEM image of Cu single atoms and cluster loaded on the graphene membrane. Atomically thin carbon adsorbates on the graphene are also observed, indicating the feasibility of graphene for the imaging of light molecules. **e** Line profiles from **d** along the red dash line, showing the image intensity of Cu single atom and graphene with high contrast. **f** EELS spectra acquired from the regions in **d**: center of Cu cluster (1), edge of Cu cluster (2) and monolayer graphene (3). The images shown in **b** and **d** are processed via Gaussian blur. Source data are provided as a Source Data file.

The intensity ratio between the O atoms and C atoms was 1.6, which matched the $\sim Z^{1.7}$ dependence of the HAADF contrast on the atomic number $Z$ of O ($Z = 8$) and C ($Z = 6$)[36]. The oxygen-containing function groups on graphene membranes not only improved their hydrophilicity, but also can efficiently anchor and stabilize the single atoms, such as Cu on graphene basal plane (Fig. 3e, f, Supplementary Fig. 17).

The hydrophilic graphene EM grids exhibit excellent stability under electron irradiation (Fig. 3g, Supplementary Fig. 18), even after irradiated with a high electron dose of $\sim$800,000 e Å$^{-2}$. The radiation robustness of graphene grid allows it to be operated in a range of operating voltage of 60–300 kV at variable temperatures. We utilized the graphene grids to load and observe biological specimens such as ferritin molecules (comprising of protein shells and iron-containing cores) at 300 kV. As shown in Fig. 3h, the ferritin molecules with an average diameter of $10 \pm 2$ nm were well dispersed on the hydrophilic graphene membrane. Energy dispersive spectrometer (EDS) element mapping of STEM indicated that ferritin cores mainly contained elements of Fe and O (Supplementary Fig. 19). The high-resolution image

showed the expected close-packed cubic lattice fringes with a lattice spacing of 2.09 Å, consistent with the spacing of the (400) planes of cubic Fe$_3$O$_4$ (Fig. 3i). Importantly, the Fe$_3$O$_4$ (440) plane spacing of 1.48 Å can also be clearly resolved at cryogenic temperature (Fig. 3j), indicating the feasibility of atomic-resolution cryo-EM imaging by using the graphene grids.

**High-resolution cryo-EM 3D reconstruction using graphene membranes.** As for the single-particle cryo-EM specimen preparation of proteins, our hydrophilic graphene grids facilitate the formation of ultrathin vitrified ice layer embedded most protein particles at graphene–water interface (Fig. 4a). We tested the specimen preparation for high-resolution cryo-EM analysis on these graphene grids using the *Thermoplasma acidophilum* 20S proteasome as an example. We found that the beam-induced motion of the 20S proteasome molecules on the graphene grids can be distinctly reduced compared to those on amorphous carbon support film (Fig. 4b). Additionally, the positions of 20S proteasomes embedded in the vitrified ice on the graphene grid were

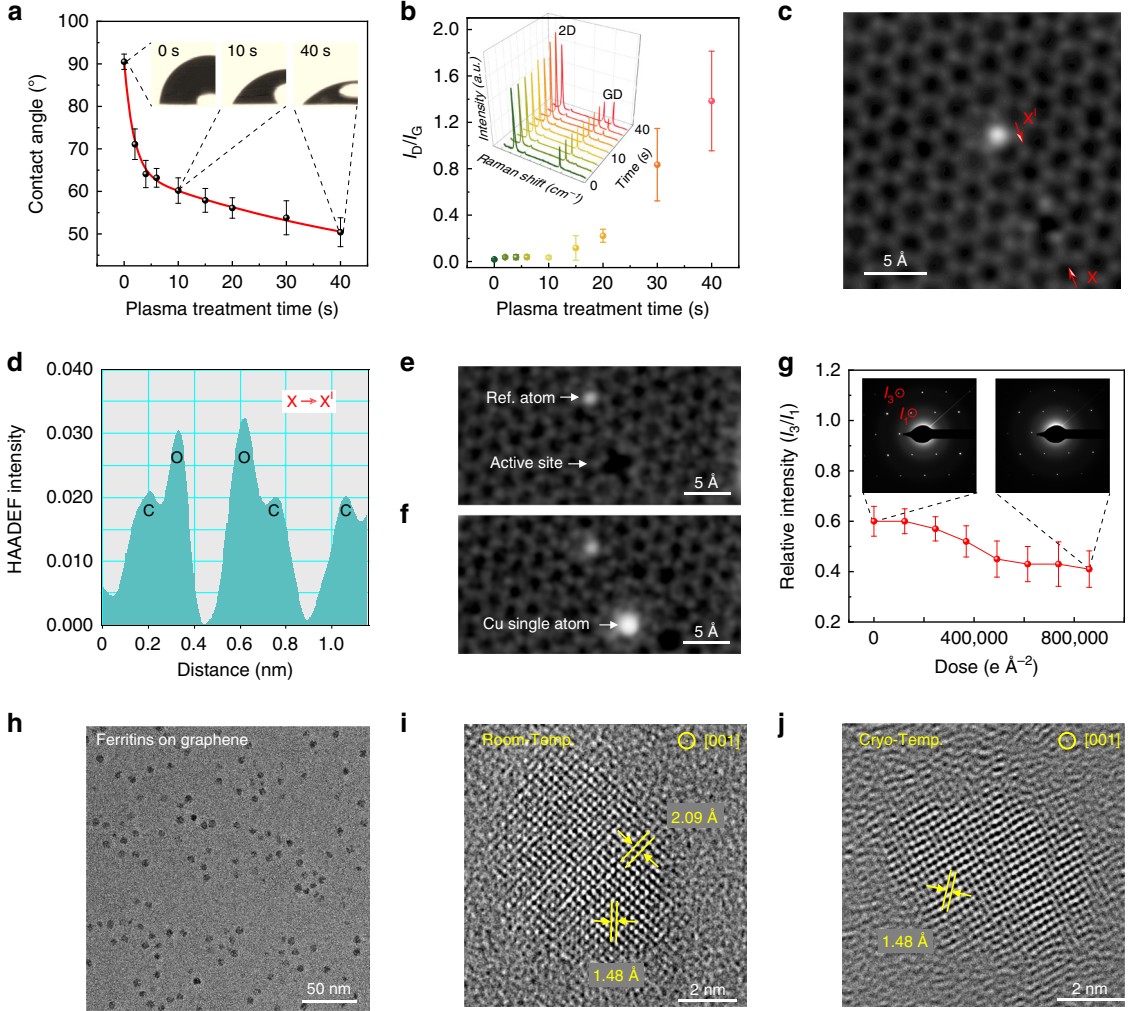

**Fig. 3 Structure and surface property manipulation of graphene membranes. a** Surface wettability manipulation of the graphene grids via a low-energy oxygen-plasma treatment. The embedded photographs are water contact angles after 0, 10 and 40 s of plasma treatments. **b** Control of surface modification on suspended graphene, indicated by intensity ratios of the D and G peaks ($I_D / I_G$) in the Raman spectra versus plasma exposure time. The inset shows the corresponding Raman spectra. **c** HAADF-STEM image of graphene after oxygen-plasma treatment. The image was corrected via Gaussian blur. **d** Line profile showing the intensity of C atoms and O atoms from **c** along X–X'. **e, f** HAADF-STEM images showing the active site on graphene surface before (**e**) and after (**f**) anchoring Cu single atom. **g** Stabilities of the plasma-treated graphene under the electron radiation, revealed by the ratio of the third-order integrated Bragg intensity ($I_3$) to the first-order ($I_1$) from the SAED patterns of graphene as a function of dose density. **h** TEM image of ferritins loaded on the graphene membrane imaged at room temperature. **i, j** Atomic-resolution TEM images of ferritin cores imaged at room and cryogenic temperatures, respectively. Source data are provided as a Source Data file.

detected by the cryo-electron tomography (cryo-ET), including the 20S proteasomes on the air–water and graphene–water interfaces (Fig. 4d). In this way, the distribution of 20S proteasomes through Z-axis can be revealed from the cryo-ET (Fig. 4c). We found that most 20S proteasomes adsorbed to the graphene–water interface (Fig. 4c, d), indicating the 20S proteasomes had strong preferences for the hydrophilic graphene surface over the air–water interface. Notably, our hydrophilic graphene grids repeatedly achieve ~30 nm thin vitrified ice layer (Supplementary Fig. 20)[37], which is suitable for cryo-EM characterization of most biological samples[38]. The electron beam radiation robustness of the supporting film in cryo-EM is another critical point. To this end, the graphene grids were continuously irradiated up to a high electron dose of 63,000 e Å$^{-2}$ at liquid-nitrogen temperature. The diffraction patterns of graphene basal plane remained sharp and the relative intensity of the Bragg reflections shows little decay during the electron irradiation (Supplementary Fig. 21), demonstrating the graphene membrane still preserved the high-quality lattice structure and proving the superior resistance to radiation.

The robust hydrophilic surface of graphene grids facilitated a uniform and high-density spatial distribution of 20S proteasomes in different orientations with fine details revealed after single-particle 2D classification and refinement (Supplementary Fig. 22). Cryo-EM and single-particle analysis from such a specimen can yield a final 3D reconstruction of the 20S proteasome at an overall resolution of 2.36 Å from ~120,000 individual molecule images (Fig. 4e). The high-resolution EM map not only allows unambiguous tracing of the backbone of the protein (Fig. 4f) but also enables distinct identification of the residue side chains (Fig. 4g), all in good agreements with previously solved atomic structures[24].

## Discussion

To summarize, we present a transfer- and polymer-free method for scalable fabrication of robust, ultraclean and hydrophilic graphene membranes. Using such graphene membranes as a specimen support, the single-atom detection and high-resolution cryo-EM is realized: the Cu single atoms can be discerned at

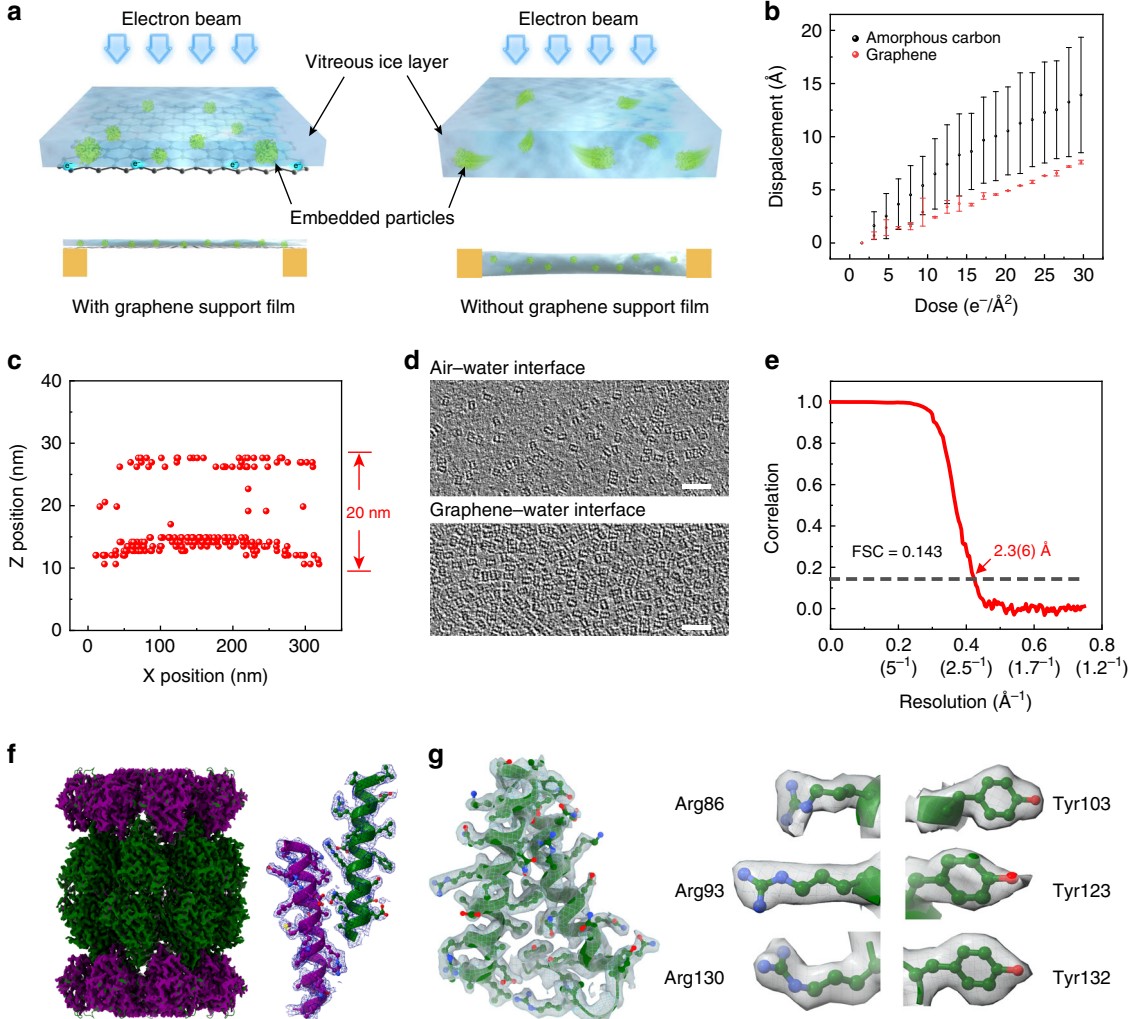

**Fig. 4 Graphene grids for high-resolution cryo-EM 3D reconstruction. a** Schematic illustration of biomolecule distributions on the graphene (left) and the porous carbon (right), wherein the graphene membrane is designed to avoid the adsorption of biomolecules to the air–water interface. **b** The protein motions under radiation on the graphene and the amorphous carbon film. Their protein motions manifest by the average displacement of 20S proteasome particles in ice relative to the initial position versus the density of electron dose. Each point represents the average displacement of thousands of particles on the graphene grids, and the error bars represent the range of values observed over different areas of the grid. **c** Positions of 20S particles on the graphene grid as indicated by cryo-ET, and each red spot represents one 20S particle. **d** Representative distribution of 20S proteasome particles on the air–water and graphene–water interfaces extracted from cryo-ET. Scale bars, 50 nm. **e** Gold-standard Fourier-shell correlation (FSC) curve for the reconstructed 20S proteasome supported on the graphene grids, where FSC = 0.143 is commonly used as a criterion to calculate the nominal resolution of cryo-EM reconstruction. **f** Overview of the 3D density map of the 20S proteasome, and detailed view of the density map for two $\alpha$-helices segmented from $\alpha$- and the $\beta$-subunits in 20S proteasome. **g** Left: close-up view of residues 81–133 from the $\alpha$-subunits with the side-chain cryo-EM density map. Right: gallery of selected residues within this region featuring side-chain densities and conformational variations. Source data are provided as a Source Data file.

atomic resolution; and a record high resolution of 2.36 Å is achieved in cryo-EM characterization of the 20S proteasome. Furthermore, besides its potential in atomic-resolution EM, robust and ultraclean suspended graphene membranes also have great potential in applications of isotope separations, gas separations and ion transports.

## Methods

**Graphene synthesis**. The clean single-crystal graphene films were grown on commercial copper foils (Alfa-Aesar #46365; a thickness of 25 μm) in a low-pressure CVD (LPCVD) tube furnace system. Firstly, the copper foil was electrochemically polished in the electrolyte solution composed of ethylene glycol and phosphoric acid (v/v = 1:3) with a voltage of 2–3 V for 10–30 min. The copper substrate was loaded into the tube furnace and heated to 1030 °C under a flow of 100 sccm $H_2$ for 1 h. The annealing of the copper substrate was carried out at 1030 °C in 100 sccm $H_2$ for 30–60 min to eliminate the surface oxide and contamination. Then the copper foil was annealed at a pressure of ~8 Pa for another 10–15 min at

1030 °C when the $H_2$ flow was shut down. This annealing treatment passivated the active sites of copper surface, which helped to suppress the nucleation density of graphene seeds and facilitated the growth of large-domain single crystalline graphene films. For the growth of graphene, 500 sccm $H_2$ and 1 sccm $CH_4$ were introduced into the LPCVD system with a pressure of ~500 Pa and maintained for 3 h. After growth, the copper foil covered with graphene films was rapidly cooled down to the desired temperature (450–550 °C, 5–10 min) under the same $H_2$ and $CH_4$ flow. Then, the supply of both $H_2$ and $CH_4$ was stopped. For the super-clean graphene preparation, $CO_2$ gas (500 sccm) was pumped into the LPCVD system to selectively remove the amorphous carbon on graphene surface at ~500 °C for 3 h, as previously reported by our group[39].

For the growth of bilayer graphene films, a piece of electrochemically polished copper foil (Alfa-Aesar #46365) was placed on a quartz substrate before loaded into the tube furnace. The copper foil was heated to 1030 °C in one hour and annealed for 30 min under a flow of 100 sccm $H_2$. Subsequently, 1000 sccm $H_2$ and 1 sccm $CH_4$ were introduced into the LPCVD system for ~1.5 h with a pressure of ~2000 Pa. Note that the high pressure of ~2000 Pa in the CVD system helps to increase the growth rate and the coverage of bilayer graphene. After growth, the bilayer graphene films were treated with $CO_2$ etching at 500 °C as mentioned above.

For the growth of few-layer graphene films, electrochemically polished Cu foil was placed on the quartz substrate and loaded into the LPCVD system. Under a flow of 100 sccm $H_2$, the copper foil was firstly heated to 1030 °C for 1 h and then annealed for 30 min. Subsequently, the graphene film was grown on the copper foil for 2 h under the flow of 2000 sccm $H_2$ and 2 sccm $CH_4$ with a pressure of ~2000 Pa. After that, the same $CO_2$ treatment was used to clean the few-layer graphene surface.

**Fabrication procedures of suspended graphene membranes.** Plasma treatment to remove the undesired graphene: To remove the graphene on one side of copper foil, we placed the copper foil with graphene on a smooth hard substrate such as clean glass slide or polyethylene glycol terephthalate (PET) film. Then the scotch tape was used to seal the edges of copper foil on the substrate. In this way, the graphene on the other side of copper foil was protected from being damaged by the following plasma treatment. After placing the hard substrate in the reactive ion etcher (Pico SLS, Diener), a flow of 10 sccm air was introduced, then the air plasma was generated at a power of 150 W for 3 min to remove the undesired graphene.

Photolithography: The positive photoresist (AR-P5350, ALLRESIST) was spun onto the Cu foil at 4000 rpm and baked at 110 °C for 3 min. Subsequently, the positive photoresist was exposed to 365 nm ultraviolet for 6–10 s with the designed mask. After being post-baked at 110 °C for 1 min, the foil was developed in dilute developer (AR 300-26, ALLRESIST; developer: water = 1:7) for 6–10 s. In this way, the patterned photoresist was fabricated on the copper foil.

Selective etching: The exposed area of the patterned copper foil was selectively etched with 0.1 M $Na_2S_2O_8$ aqueous solution. Note that a stir bar could be used for controlling shapes of the holes.

Rinsing: The floating graphene grids were washed gently with deionized water to remove etchant traces, and the photoresist was carefully removed with *N*-methyl pyrrolidone and acetone; then the grids were rinsed in the isopropyl alcohol.

Drying: The bilayer and few-layer graphene grids were normally dried in the super-clean room to avoid extra contaminants after being took out from the isopropyl alcohol. The monolayer graphene grids were dried in the critical point drier (samdri-795) to protect the suspended graphene membranes from rupturing by surface tension.

**Controllable oxygen-plasma treatment.** A reactive ion etcher (Pico SLS, Diener) was used to modulate the species of source gas, volume flow rate of gas, treatment time and power (Supplementary Fig. 23). For controllable oxygen-plasma treatment, graphene grids were placed directly on a metal substrate in the plasma chamber. Then the oxygen plasma was generated with a low-energy power of 40 W and a flow of 5 sccm oxygen. By controlling the plasma treatment time, desirable wettability and defect density of graphene grid can be achieved (Fig. 3a, b).

**Characterization of graphene membranes.** The morphology and property of graphene membranes were characterized by optical microscopy (OM) (Nikon, DS-Ri2), scanning electron microscopy (SEM) (acceleration voltage 1–2 kV, Hitachi S-4800) and Raman Spectrometer (514 nm, LabRAM HR-800, Horiba). The WCAs of graphene grids were measured using contact angle measurement (Dataphysics OCA 20). The selected area electron diffraction (SAED) patterns of graphene were collected by transmission electron microscopy (TEM, FEI Tecnai F30, acceleration voltage 300 kV). And the aberration-corrected STEM images of graphene were performed using a Nion U-HERMS200 microscope at 60 kV.

**Characterization of single atoms on suspended graphene membranes.** The aberration-corrected STEM images and electron energy loss spectrum (EELS) data were performed using a monochromatic Nion U-HERMS200 electron microscope which is operated at 60 kV. Both aberration-corrected STEM images and EEL spectra were acquired with a 35-mrad convergence semi-angle. The images were acquired using a HAADF detector with an 80–210 mrad collection semi-angle and the EEL spectra data were acquired with a 24.9-mrad collection semi-angle. EEL spectra were recorded by the EELS CCD with 2048 channels, the dispersion of which is 0.41 eV/channel, hence we could detect the loss range of 200–1050 eV which contains the K edge of carbon (284 eV) and L edge of copper (931 eV). The diameter of the focused electron beam is at the angstrom level. To minimize the damage of such a focused beam and acquire signals from multi atoms, we made the beam slightly defocused of 50 nm and set the exposure time as 5 s for each spectrum.

**TEM characterization of ferritins on suspended graphene membranes.** The ferritins from equine spleen (Sigma-Aldrich) were used for TEM characterization. Specifically, 1 μL of ferritin solution diluted 100 times from the original solution was loaded onto the membrane surface at room temperature, followed by drying naturally in the air. At room temperature, TEM characterizations and the EDX mapping of ferritins were carried out using a FEI 80–300 Environmental Titan operated in monochromated mode at 300 kV.

For the characterization at cryogenic temperature, the as-prepared graphene grid with ferritin was plunged-frozen into liquid nitrogen and then carefully mounted onto a TEM cryo-holder (Gatan 626 cryo-holder) using a cryo-transfer station to ensure the entire process under liquid $N_2$. The cryo-holder is then inserted into the TEM column (~1 s), and during the whole process the sample is kept at −178 °C. TEM characterizations were carried out using a FEI Titan 80-300 environmental (scanning) transmission electron microscope (E(S)TEM) operated at 300 kV. The microscope was equipped with an aberration corrector in the image-forming (objective) lens, which was tuned before each sample analysis.

**Vitrification and cryo-EM with graphene membranes.** The *T. acidophilum* 20S proteasome was provided by the Xueming Li lab in Tsinghua University[24]. Shortly, His-tagged 20S plasmids were transformed into *Escherichia coli* cells and induced for protein expression. The protein complex was purified by nickel affinity columns as previously reported. For cryo-EM specimen preparation, graphene grids were glow-discharged for 10 s, and then ~3 μL of 20S proteasome solution at a concentration of ~0.2 mg mL$^{-1}$ was loaded onto the grids. After incubation for 2 s in a cryo-plunger at 8 °C and 100% relative humidity (FEI Vitrobot, Thermo Fisher Scientific Inc.), the grids were blotted for 5 s with filter paper and plunge-frozen into liquid ethane cooled at liquid nitrogen temperature. The grids were stored in liquid nitrogen until examined by cryo-EM.

The as-prepared frozen specimen grids were loaded into a FEI Titan Krios (300 kV) TEM equipped with an X-FEG and Cs-corrector, and the dataset was collected by Gatan K2 direct electron detector camera (Gatan Company). A total of 935 micrographs were recorded by AutoEMation2 (developed by Jianlin Lei, Tsinghua University), with a pixel size of 0.67 Å and defocus range of 0.5–1.9 μm. Each micrograph was fractionated to 32 frames with a collective dose of 50 e Å$^{-2}$ and these frames were motion-corrected by MotionCor2 (REF). CTF estimation was performed by CTFFIND4 and 156667 particles were auto-picked by Relion (REF). After 2D and 3D classification, 124,212 particles were finally used for further 3D auto-refine with D7 symmetry. The resolution of the final 3D reconstruction was reported as 2.3(6) Å estimated by the FSC 0.143 criterion (Fig. 4f). The cryo-EM structure of 20S proteasome (PDB ID 3J9I) was fit into the map with UCSF Chimera's rigid body-fitting algorithm and refined with phenix.real_space_refine[40] and manually adjusted in Coot[41] iteratively. The 3D renderings of the maps and models were created in UCSF-Chimera (REF).

**Particles movement measurement.** To measure the displacement of particles on various supporting membranes, we collected movies of specimens under Titan Krios (300 kV) with constant dose rate (with accumulated dose of ~50 e Å$^{-2}$), and each micrograph was dose-dependently fractionated to 32 frames. Particles were auto-picked and their coordinates in each frame were determined by Relion3.0. These coordinates were then applied for calculating the displacements of 20S proteasome particles on individual supporting membrane versus dose accumulation.

**Stability test of graphene membranes.** The stability of graphene membranes at liquid nitrogen temperature was investigated by Tecnai F20 (200 kV). Selected area electron-diffraction patterns of graphene were continuously collected at the same position on a CCD camera (Gatan US4000) with a dose rate of ~15 e Å$^{-2}$ s$^{-1}$ from 0 min to 70 min. The relative integrated intensity variation of diffractograms (the first-order integrated intensity was divided by the third-order integrated intensity) versus dose was calculated to evaluate the stability of the graphene membranes.

## Data availability
Data supporting the findings in this manuscript are available from the corresponding authors upon reasonable requests. The source data underlying Figs. 1f, 2b, 3a, b, 3g and 4b and Supplementary Figs. 11, 12, 16, 20 and 21 are provided as a Source Data file.

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

## Acknowledgements

We thank Z.B. Liang for technical assistance in critical-point drying; X. Fan and Z.P. Luo for facilitating cryo-EM characterization; S.L. Chen, S.S. Zhang and T. Wang for the help in structural characterization; J.X. Wu for fruitful discussions and comments. We thank the Electron Microscopy Laboratory at Peking University for the use of electron microscopes. We acknowledge financial support from the National Basic Research Program of China (No. 2016YFA0200101), the National Natural Science Foundation of China (Nos. 21525310, 31825009, 51672007, 11974023, 91963113, 11772003 and 11890681), Ministry of Science and Technology of China (grant number 2016YFA0501101), Key Area R&D Program of Guangdong Province (2018B010109009 and 2018B030327001).

## Author contributions

L.Z., N.Li, J.Z. and W.D. carried out the synthesis, structural characterization and fabrication of the suspended graphene membranes. B.D., L.S. and H.Y. contributed to the growth of large-area graphene membranes on the copper foil. C.T. characterized the large-area graphene grids. Y.C., N.Liu, Z.Y., Y.L., Y.C. and H.-W.W. contributed to the cryo-EM characterizations and analysis. L.Z., N.Li and P.G. performed the characterization of single atoms. L.Z., H.P., X.G., J.L. and X.W. contributed to the mechanical analyses. H.P., L.Z., M.W., S.X. and Y.C. wrote/corrected the manuscript. H.P. conceived and supervised the project. All authors discussed the results and commented on the manuscript.

## Competing interests

The authors declare no competing interests.
