## [Peer Review File · Nature Communications]

Reviewers' comments:

Reviewer #1 (Remarks to the Author):

This manuscript reports a novel way to make ultraclean, robust graphene membranes that span the open spaces in EM grids. This innovation will be of great interest to all who are developing better methods to prepare specimens for cryo-EM.

The manuscript also reports spectroscopic measurements, high-resolution imaging achievements, and characterization of the consequences of oxygen plasma treatment of the graphene membranes. These additional elements are not well motivated within the context of the main innovation, however. I thus recommend a major revision in which work peripheral to the main innovation is removed. In addition, I recommend the addition of substantially more detail about a several aspects of the main innovation.

The polymer- and transfer-free direct-etching method reported here will be of real interest to others. Thus, in addition to the information already provided, I recommend that more be provided about the following points.

1. The authors mention in lines 66 and 67 of page 3, and in the legend for Figure 4, that graphene support films can be used to avoid adsorption of proteins to the air-water interface. Graphene itself should be made more hydrophilic, however, and the authors should cite other papers in addition to reference 11 on this point. Russo, for example, recently advocated exposure to a helium plasma into which volatile molecules with different types of chemical functionality were introduced, and Kuehlbrandt adsorbed pyrene carboxylic acid to the surface of graphene.
2. Please mention the thickness of the EM grids that are made in this process; perhaps state, in line 121 on page 5, the thickness of the copper foil that was used.
3. Please provide more detail about how graphene was removed from the "back" side of the copper foil. How was it confirmed that graphene was removed without damaging the graphene on the "front" side? Please confirm that removal of graphene from the back side is actually necessary, or whether it simply seems the right thing to do.
4. Please provide all information that is thought to be relevant about the "controllable oxygen-plasma treatment" used here. Among other things, were the grids placed directly in the plasma or were they downstream in the flow of gas that still contains reactive-oxygen species?
5. Add further information about the number of oxygen atoms added per unit area of the graphene as a result of the oxygen plasma treatment. How well does the count of oxygen atoms (per unit area) that are seen in HAADF images compare to the chemical composition determined by other methods, such as EELS, EDS, or XAFS, if such a measurement was performed? Is there already some oxygen adduct even before treatment with an oxygen plasma?
6. The statement is made in lines 180 and 181 of page 8 that "our hydrophilic graphene grids facilitate the formation of ultrathin vitrified ice layer embedded ... protein particles." Please specify the type of grid to which these graphene grids are being compared. Is it grids with the same graphene, before exposure to an oxygen plasma? Or is it a more general claim that the treated graphene facilitates the formation of ultrathin vitrified ice even better than does the use holey carbon?
7. Figure 4c shows a representation of particle positions derived from electron tomography, in an area where the ice is less than 20 nm thick. Please provide documentation about how common it is to find such thin areas. It would be appropriate to show a grid-atlas image, and documentation about how much variability there is in the appearance of such atlases.
8. Have the authors considered using an energy filter to measure the percent of inelastically scattered electrons over different areas of the grid, and thus to easily measure the ice thickness values? (Rice, W. J., A. C. Cheng, A. J. Noble, E. T. Eng, L. Y. Kim, B. Carragher, and C. S. Potter. 2018. Routine determination of ice thickness for cryo-EM grids. *Journal of Structural Biology* 204:38-44.)
9. It is stated, in lines 184 – 186 of page 8, that beam-induced motion is distinctly reduced. This outcome might not have been expected, however, since (a) graphene is expected to "crinkle" at

least as much as holey carbon film when cooled to nitrogen temperature (due to differential thermal contraction of copper grids vs carbon support films; Booy, F. P., and J. B. Pawley. 1993. Cryo-Crinkling - What Happens to Carbon-Films on Copper Grids at Low-Temperature. Ultramicroscopy 48:273-280), (b) single-atom thick, or even few-layer thick graphene is expected to be even more flexible than 10 nm or 20 nm thick holey carbon, and (c) doming is expected to be greater over a 30 – 50 micrometer opening than over a 1 – 2 micrometer hole. Since the outcome shown in Figure 4b is thus opposite to what might be expected, strong statistical evidence should be presented in order to document that the result is consistently the case. It is well known that holey carbon films show considerable variation with respect to the magnitude of beam-induced motion, and thus it would be important to show whether the same is true for the new, graphene grids.

10. Lines 187 and 188 mention that 20S proteasomes have a preference for the graphene interface over the air-water interface. Since supplementary Figure 19 b indicates that particles exhibit mainly side views, there may also be a preferred-orientation issue. More information should be provided on this question. In particular, please clarify whether the word “uniform” that is used in line 197 was meant to refer to spatial distribution or to angular distribution. Also please keep in mind that preferential orientation may not result in poor resolution of the 3-D map because of the high point-group symmetry of the proteasome. Readers will be very much interested in whether the cylinder-shaped particles are bound to graphene in a highly preferred orientation.

11. Some details are given of conditions used to grow large, single-crystal graphene, including the fact that methane was used as the feedstock. Since the authors emphasize in the title of the manuscript that the graphene is ultraclean, does this mean that the novel, polymer- and transfer-free method described here is sufficient to produce ultraclean graphene, and that copper acetate is not necessary? It would seem to be important for the authors to cite some of recent publications in which copper acetate is used as the feedstock, and to comment on the relative merits of the two choices for making ultraclean graphene.

12. Figure 1 f shows data on the “intactness” statistics. Please confirm whether this is for grids to which no sample had yet been loaded. If so, please add information on how these statistics change after sample is loaded, and, in particular, please add information of how these statistics change when the grids are used to prepare cryo-EM samples.

Reviewer #2 (Remarks to the Author):

Liming Zheng et al present a newly developed and detailed protocol to fabricate ultraclean graphene grids for high-resolution cryo-EM. The importance of graphene grid supports in achieving high resolution structures have been previously shown and sparked a large interest in the cryo-EM community. Despite encouraging initial reports, a reliable and reproducible protocol to yield high-quality grids has been proven extremely challenging.

This study provides evidences that a robust protocol for stable graphene membranes with low background noise and tunable surface properties has been identified. Moreover, the quality and reliability of such grids has been successfully tested by solving the 20S proteasome structure to a remarkable 2.36 Å in resolution.

Together, the study provides a scalable method for graphene grids fabrication that, once available to the cryo-EM community, will facilitate specimen vitrification and improve image quality during cryo-EM data collection.

The manuscript is well written with detailed results that support the conclusion made by the authors.

Reviewer #3 (Remarks to the Author):

In my opinion, the manuscript of Zheng et al. "Robust ultraclean atomically thin membranes for atomic-resolution electron microscopy", despite being quite obvious (even I have tried the same approach for preparing ultraclean graphene for TEM grids) is interesting enough for a broad audience and can be eventually published in Nature Communications but only after major revision. The authors demonstrate the advantages of graphene based membrane grids produced using a polymer- and transfer-free direct-etching method. The resulting graphene membranes are robust, very clean and can be tuned to have different surface hydrophilicity. Demonstrating these advantages in evaluation of actual samples improves the paper and illustrates how negligible background noise and good stability under electron irradiation allows imaging with atomic-resolution including proteins at the graphene-ice interface.

The weaknesses that need to be fixed include poorly written Methods section, especially the details of graphene synthesis, plasma irradiation for hydrophilic modification and others. Some sentences are not even finished, like on line 225. There are also multiple language flaws that could be fixed by a native speaker.

On the scientific part, I have the following comments:

1. The use of Young-Laplace equation $P_{cap} = 2\gamma_l - g/RI$ on page 102 for calculation of the capillary pressure P_{cap} is not supported by illustration of RI on Figure 1.
2. It would be helpful to show the survival yield variation with the hole diameter in Figure 1f or Table S1. It is also not clear when normal and when the supercritical drying were used.
3. Some data in Figures are not mentioned or poorly described in the text, e.g. reference to Figure 4d is unclear and no definition is given to Gold-standard Fourier-shell correlation (FSC) in Figure 4e

Response to the 1st Reviewer

This manuscript reports a novel way to make ultraclean, robust graphene membranes that span the open spaces in EM grids. This innovation will be of great interest to all who are developing better methods to prepare specimens for cryo-EM.

The manuscript also reports spectroscopic measurements, high-resolution imaging achievements, and characterization of the consequences of oxygen plasma treatment of the graphene membranes. These additional elements are not well motivated within the context of the main innovation, however. I thus recommend a major revision in which work peripheral to the main innovation is removed. In addition, I recommend the addition of substantially more detail about a several aspects of the main innovation.

The polymer- and transfer-free direct-etching method reported here will be of real interest to others. Thus, in addition to the information already provided, I recommend that more be provided about the following points.

Response:

We appreciate very much the positive comments and constructive suggestions from the reviewer on the main innovation of our work. To evaluate the high quality of graphene membranes fabricated by our polymer-free and transfer-free process, we think the spectroscopic measurements and high-resolution imaging were necessary to demonstrate the clean surface of suspended graphene, which is both important in the cryo-EM and materials science (Fan X. *et al. Nat. Commun.* 2019, 10, 2386; Lin L. *et al. Chem. Rev.* 2018, 118, 9281). Considering that the chemical structures and functional groups on the functionalized graphene films are still not well characterized (Glaeser, R. M. *Curr. Opin. Colloid. In.* 2018, 34, 1), it is essential to use the high-resolution imaging and spectroscopy measurements to characterize the structure of oxygen-containing functional groups on the plasma-treated graphene, helping to illustrate the relationship between the structure and hydrophilicity of the functionalized graphene. We hope these complete characterizations could make our manuscript more understandable and informative to readers from the interdisciplinary research fields (materials science, cryo-EM, nanochemistry, nanotechnology, ...).

To strengthen the presentation of our paper, we will fully address the reviewer's comments and concerns point by point in the following.

1. The authors mention in lines 66 and 67 of page 3, and in the legend for Figure 4, that graphene support films can be used to avoid adsorption of proteins to the air-water interface. Graphene itself should be made more hydrophilic, however, and the authors should cite other papers in addition to reference 11 on this point. Russo, for example,

recently advocated exposure to a helium plasma into which volatile molecules with different types of chemical functionality were introduced, and Kuehlbrandt adsorbed pyrene carboxylic acid to the surface of graphene.

Response:

We thank the referee for the valuable suggestion. The hydrophilic graphene membranes facilitate the efficient loading of proteins from aqueous solutions. The helium plasma treatment and the adsorption of pyrene carboxylic acid are effective ways to improve the hydrophilicity of the graphene, and we have cited these papers and revised the manuscript as suggested.

2. Please mention the thickness of the EM grids that are made in this process; perhaps state, in line 121 on page 5, the thickness of the copper foil that was used.

Response:

Thanks for the reviewer's suggestions. We have accordingly stated the thickness of the copper foil in the main text of the manuscript as follows: "...we successfully fabricated graphene EM grids in an array as large as 18×18 grids in counts from a 64 cm² Cu foil with a thickness of ~25 μm..."

3. Please provide more detail about how graphene was removed from the "back" side of the copper foil. How was it confirmed that graphene was removed without damaging the graphene on the "front" side? Please confirm that removal of graphene from the back side is actually necessary, or whether it simply seems the right thing to do.

Response:

We thank the referee to raise this concern. To remove the graphene from the "back" side of copper foil, we placed the copper foil with graphene on a smooth hard substrate such as clean glass slide or polyethylene glycol terephthalate (PET) film. Then the scotch tape was used to seal the edges of copper foil on the substrate (Response Figure 1a). In this way, the graphene on the "front" side was protected from being damaged by the following plasma treatment. After placing the hard substrate in the reactive ion etcher (Pico SLS, Diener), a flow of 10 sccm air was introduced, then the air plasma was generated at a power of 150 W for 3 min to remove the undesired graphene.

After air plasma treatment, the quality of the protected graphene was evaluated by the Raman spectroscopic measurements. Raman spectroscopy has been widely used to investigate the structure disorder in the sp^2 -network of graphene. The Raman signature for structure disorder appears at ~1350 cm⁻¹ called D band, and the intensity ratio of the

D and G bands (I_D/I_G) is commonly used to evaluate the quality of graphene (Ferrari, A. C. *et al. Nano Lett.* 2011, 11, 3190). In Response Figure 1b, no D band was observed for any of the suspended graphene membranes, indicating the high quality of the protected graphene film. In addition, the atomic-resolution images of graphene in Figure 2b also showed the intact structure of the protected graphene. Therefore, we can conclude that the undesired graphene can be removed without damaging the graphene on the “front” side.

If we don't remove the graphene on the back side, the released graphene would roll up into scrolls under the liquid surface tension after etching the copper substrate. And the scrolls subsequently attached onto the remaining top graphene film (Response Figure 1c), corresponding with the previous reported result (Bao, Z. *et al. Sci. Adv.* 2017, 3, 1700159). Actually, once the undesired graphene films were not removed totally, the graphene flake residues would pollute the suspended graphene membranes (Response Figure 1d). Thus, it's better to remove the graphene from one side of the copper foil.

Response Figure 1. Removal of graphene on one side of the copper foil. a, Schematic illustration of copper foil with graphene sealed by the scotch tape on the substrate. b, Representative Raman spectra from one graphene grid. c, Optical image of graphene film with scrolls on the SiO₂/Si substrate. d, SEM image of suspended graphene membranes, the red arrow shows that one piece of graphene flake attaches on the suspended graphene.

To make removal of graphene more straightforward to readers, we have accordingly revised the manuscript as follows: “... Plasma treatment to remove the undesired graphene: To remove the graphene on one side of copper foil, we placed the copper foil with graphene on a smooth hard substrate such as clean glass slide or polyethylene glycol terephthalate (PET) film. Then the scotch tape was used to seal the edges of copper foil on the substrate. In this way, the graphene on the other side of copper foil was protected from being damaged by the following plasma treatment. After placing the hard substrate in the reactive ion etcher (Pico SLS, Diener), a flow of 10 sccm air was introduced, then the air plasma was generated at a power of 150 W for 3 min to remove the undesired graphene...”

4. Please provide all information that is thought to be relevant about the “controllable oxygen-plasma treatment” used here. Among other things, were the grids placed directly in the plasma or were they downstream in the flow of gas that still contains reactive-oxygen species?

Response:

We thank the referee to raise this concern. We can modulate the species of source gas, volume flow rate of gas, plasma treatment time and power with the reactive ion etcher (Pico SLS, Diener) (Response Figure 2). For the controllable oxygen-plasma treatment, the graphene grids were placed directly on a metal substrate in the plasma chamber, and we controlled the power at a low energy of 40W and the volume flow rate at a flow of 5 sccm oxygen. By regulating the time of oxygen plasma treatment, desirable wettability and surface modification of graphene grid can be achieved (Figure 3a-b).

Response Figure 2. Reactive ion etcher. The chamber of reactive ion etcher and the control systems of source gas, volume flow rate, time and power are marked in the optical image, respectively.

To make the controllable oxygen-plasma treatment clearer to readers, we have accordingly revised the manuscript as follows: “... The reactive ion etcher (Pico SLS, Diener) used here can modulate the species of source gas, volume flow rate of gas, plasma treatment time and power. For the controllable oxygen-plasma treatment, the graphene grids were placed directly on a metal substrate in the plasma chamber, then the power was controlled at a low energy of 40W and the volume flow rate was manipulated at a flow of 5 sccm oxygen. By regulating the time of oxygen plasma treatment, desirable wettability and surface modification of graphene grid can be achieved...”

5. Add further information about the number of oxygen atoms added per unit area of the graphene as a result of the oxygen plasma treatment. How well does the count of oxygen atoms (per unit area) that are seen in HAADF images compare to the chemical composition determined by other methods, such as EELS, EDS, or XAFS, if such a measurement was performed? Is there already some oxygen adduct even before treatment with an oxygen plasma?

Response:

We thank the referee for the constructive comments. Quantifying the active sites in graphene is significant to get insight into their fundamental properties and further applications. Raman spectroscopy is powerful in quantifying the defects in graphene. (Ferrari, A. C. *et al. Nano Lett.* 2011, 11, 3190; Araujo, P. T. *et al. Mater Today* 2012, 15, 98) Typically, the defect density σ can be described by the average distance between defects L_D , where $\sigma=1/L_D^2$. And the L_D can be calculated from the intensity ratio of the D and G bands (I_D/I_G) in Raman spectra. (Lucchese, M. M. *et al. Carbon* 2010, 48, 1592) In the low defect density regime ($L_D > 6$ nm),

$$I_D/I_G = C/L_D^2;$$

and in the high defect density regime,

$$I_D/I_G = D \cdot L_D^2.$$

For the oxygen-plasma induced defects in graphene, the C is reported to be 102 nm² and D is obtained by imposing continuity between the two regions. (Ardavan, Z. *et al. Nat. Commun.* 2014, 5, 3186; Childres, I. *et al. New J. Phys.* 2011, 13, 025008; Lucchese, M. M. *et al. Carbon* 2010, 48, 1592) In our cases, the maximum I_D/I_G is ~1.8 (Figure 3b) and the corresponding L_D is ~8 nm ($L_D > 6$ nm), revealing the low defect density in the functional graphene. Thus, we can calculate the average distance between defects (L_D) using

$$I_D/I_G = 102/L_D^2.$$

Supplementary Figure 20. Quantifying the site-defect density in plasma-treated graphene. a, The variable intensity ratios of the D and G peaks (I_D / I_G) with corresponding average distance between defects (L_D). Not that L_D is plotted as top x axis, and L_D is showed at the corresponding oxygen plasma treatment time without any linear relationship. The values of oxygen-containing active site density σ from 0 s to 40 s are calculated to be $1.76 \times 10^{-4} \text{ nm}^{-2}$, $3.64 \times 10^{-4} \text{ nm}^{-2}$, $2.18 \times 10^{-3} \text{ nm}^{-2}$, $8.12 \times 10^{-3} \text{ nm}^{-2}$ and $1.35 \times 10^{-2} \text{ nm}^{-2}$ using $\sigma = 1/L_D^2$, respectively. b-f, High-resolution TEM images of defects in the graphene with different plasma treatment time: 0 s (b), 10 s (c), 20 s (d), 30 s (e), 40 s (f). The black dots in red circles reveal the point defects that correspond to the oxygen adducts (Ardavan Z., *et al. Nat. Commun.* 2014, 5, 3186); and the distances between defects are marked with the white arrows. Scale bars, 5 nm.

The calculated L_D with corresponding I_D/I_G intensity ratio was shown in the Supplementary Figure 20a. During the initial 10 s plasma exposure time, the I_D/I_G was largely suppressed (~ 0.05), indicating the high quality of suspended graphene membranes. This was demonstrated by the high-resolution TEM (HRTEM) images of graphene, where no obvious site defect was observed (Supplementary Fig. 20b-c). As the exposure time was increased to 20 s, the average L_D was ~ 20 nm, corresponded with the HRTEM observations (Supplementary Fig. 20d). The average L_D further decreased to ~ 10 nm when the exposure time prolonged to 30~40 s (Supplementary Fig. 20a, e-f). The values of oxygen-containing active site density σ from 0 s to 40 s are calculated to be $1.76 \times 10^{-4} \text{ nm}^{-2}$, $3.64 \times 10^{-4} \text{ nm}^{-2}$, $2.18 \times 10^{-3} \text{ nm}^{-2}$, $8.12 \times 10^{-3} \text{ nm}^{-2}$ and $1.35 \times 10^{-2} \text{ nm}^{-2}$ using $\sigma = 1/L_D^2$,

respectively. Considering that the spot size of laser is $\sim 1 \mu\text{m}$ for the $100\times$ objective in a Raman spectrometer, the detection area of Raman spectroscopy is much larger than that of HRTEM. Thus, the Raman spectroscopic measurements are more statistically significant to quantify the site-defect density in graphene.

Response Figure 3. Ultraclean graphene before oxygen-plasma treatment. a, HADDF-STEM image of ultraclean graphene. The continuous clean size was 100 nm without any oxygen adduct. b, Typical HADDF-STEM atomic-resolution image of ultraclean graphene regions as marked in a. c-d, High-magnification HADDF-STEM images of the graphene, showing the hexagonal arrangement of the carbon atoms in graphene without any impurities and defects. The images in b-d were processed via Gaussian blur.

Furthermore, we performed the HADDF-STEM characterizations to investigate whether oxygen adduct existed in graphene lattice before plasma treatment. As shown in Response Figure 3, no oxygen adducts or defects were observed in graphene lattices, demonstrating the intact structure of graphene before plasma treatment. In addition, no D band was detected using Raman spectroscopy before the graphene membranes were functionalized (Figure 3b), revealing that no oxygen adducts existed in the sp^2 -network of graphene. Conversely, once the oxygen adducts were introduced in the graphene lattices, the structure disorder in high-resolution images (Fig. 3c, Supplementary Fig. 20) and the obvious D bands in Raman spectra would be easily observed. Therefore, we tend to believe that no oxygen adducts exist in the graphene lattice before oxygen plasma treatment.

To make the control and quantification of site-defect density more straightforward to readers, we have accordingly revised the manuscript as follows: “... Meanwhile, the site-defect density of graphene and the average distance between defects (L_D) can be modulated via adjusting the condition of plasma treatments. As shown in Figure 3b, the intensity ratio of D to G peak (I_D/I_G) of Raman spectroscopy for graphene membrane maintained suppressed (~ 0.05) within 10 s of plasma treatments, and then I_D/I_G distinctly increased to ~ 2 with a corresponding L_D of ~ 10 nm as the treatment prolonged (Supplementary Fig. 20)...” In addition, we have added one figure (Supplementary Figure 20) in supporting information.

6. The statement is made in lines 180 and 181 of page 8 that “our hydrophilic graphene grids facilitate the formation of ultrathin vitrified ice layer embedded ... protein particles.” Please specify the type of grid to which these graphene grids are being compared. Is it grids with the same graphene, before exposure to an oxygen plasma? Or is it a more general claim that the treated graphene facilitates the formation of ultrathin vitrified ice even better than does the use holey carbon?

Response:

We thank the referee to raise this concern. We counted the ice layer thickness of more than 100 locations, and found that the thickness centered at ~ 30 nm, as shown in Supplementary Figure 21. This ice layer thickness is suitable for cryo-EM characterization of most biological samples. (Glaeser, R. M. *Curr. Opin. Colloid. In.* 2018, 34, 1) We think that the ice layer on graphene is generally thinner than that of holey carbon (~ 50 nm). (Liu, N. *et al. J. Am. Chem. Soc.* 2019, 141, 4016) To make the claim more rigorous, we have modified the description in the revised manuscript:

“...Such grids have also been proved to achieve ultrathin vitrified ice layer (~~20-30~~ ~ 30 nm in thickness) with most protein particles surrounded at the graphene-ice interface...”

“...our hydrophilic graphene grids repeatedly achieve ~ 30 nm thin vitrified ice layer (Supplementary Fig. 21), which is suitable for cryo-EM characterization of most biological samples...” In addition, we have added one figure (Supplementary Figure 21) in supporting information.

Supplementary Figure 21. Statistics of ice layer thickness from more than 100 locations.

7. Figure 4c shows a representation of particle positions derived from electron tomography, in an area where the ice is less than 20 nm thick. Please provide documentation about how common it is to find such thin areas. It would be appropriate to show a grid-atlas image, and documentation about how much variability there is in the appearance of such atlases.

Response:

We thank the referee to raise this concern. We have added the grid-atlas image, as shown in Response Figure 4. We have made statistics on the ice thickness variation of the atlas and the thickness varied from ~ 20 nm to ~ 60 nm (Supplementary Fig. 21).

Response Figure 4. a, The grid-atlas image. b, The representatively enlarged image. The red circle refers to the location where data was normally collected.

8. Have the authors considered using an energy filter to measure the percent of inelastically scattered electrons over different areas of the grid, and thus to easily measure the ice thickness values? (Rice, W. J., A. C. Cheng, A. J. Noble, E. T. Eng, L. Y.

Kim, B. Carragher, and C. S. Potter. 2018. Routine determination of ice thickness for cryo-EM grids. *Journal of Structural Biology* 204:38-44.)

Response:

Thanks for your suggestion. We followed the recommendation, and used the energy filter to measure the ice thickness (Supplementary Fig. 21). The measurement method refers to references: Rice, W. J. *et al. Journal of Structural Biology* 2018, 204, 38; Cho, H.J. *et al. Journal of Analytical Science and Technology* 2013, 4, 7.

9. It is stated, in lines 184 – 186 of page 8, that beam-induced motion is distinctly reduced. This outcome might not have been expected, however, since (a) graphene is expected to “crinkle” at least as much as holey carbon film when cooled to nitrogen temperature (due to differential thermal contraction of copper grids vs carbon support films; Booy, F. P., and J. B. Pawley. 1993. Cryo-Crinkling - What Happens to Carbon-Films on Copper Grids at Low-Temperature. *Ultramicroscopy* 48:273-280), (b) single-atom thick, or even few-layer thick graphene is expected to be even more flexible than 10 nm or 20 nm thick holey carbon, and (c) doming is expected to be greater over a 30–50 micrometer opening than over a 1 – 2 micrometer hole. Since the outcome shown in Figure 4b is thus opposite to what might be expected, strong statistical evidence should be presented in order to document that the result is consistently the case. It is well known that holey carbon films show considerable variation with respect to the magnitude of beam-induced motion, and thus it would be important to show whether the same is true for the new, graphene grids.

Response:

We thank the referee to raise this concern. Owing to the differential thermal contraction of copper grid and carbon support films, carbon shrinks less than the copper when cooled to the liquid-nitrogen temperature. This leads to the “cryo-crinkling” of amorphous carbon film and results in the increased beam-induced motion. (Glaeser, R. M. *Ultramicroscopy* 1992, 46, 33; Booy F.P. *et al. Ultramicroscopy* 2009 48, 273; Glaeser, R. M. *et al. Ultramicroscopy* 2011, 111, 90-100; Russo C.J., *et al. J. Struct. Biol.* 2016, 193, 33) However, the use of thicker amorphous carbon film can reduce the motion, which is probably due to the enhanced conductivity and increased mechanical strength (Glaeser, R. M. *et al. Ultramicroscopy* 2011, 111, 90).

Our observations may also provide an insight into the role of “conductivity” in the reduction of beam-induced motion. The thermal expansion mismatch between the graphene ($\alpha_{Gr} = -7 \times 10^{-6} / K$) and copper substrate ($\alpha_{Cu} = -16.6 \times 10^{-6} / K$) is about ~0.5%, when the graphene grid is cooled from room temperature to the liquid-nitrogen

temperature. (Deng B. *et al. ACS Nano* 2017, 11, 12337) The mismatch is comparable with that of amorphous carbon film on the copper grid ($\sim 0.3\%$). (Glaeser, R. M. *et al. Ultramicroscopy* 2011, 111, 90) Accordingly, we observed the similar “cryo-crinkling” in suspended graphene as the referee expected (Response Fig. 5a). However, the beam-induced motion of proteins on our suspended graphene is still statistically reduced in contrast to that on the amorphous carbon (Response Fig. 5b). And the amorphous carbon film shows a much larger variation than the suspended graphene, which is consistent with the previous reported results (Russo C.J. *et al. Nat. Methods.* 2014, 11, 649).

Considering that the amorphous carbon film (~ 5 nm) is much thicker than the graphene membrane (< 1 nm), there would be more inelastic and multiple scattering effects that degrade the image quality for the amorphous carbon film. (Russo C.J. *et al. Current Opinion in Structural Biology* 2016, 37, 81) In addition, the thermal conductivity of graphene (> 5000 W mK $^{-1}$) is over 3 orders of magnitude higher than that of amorphous carbon (~ 0.01 -1 W mK $^{-1}$); and the electrical resistivity of graphene ($\sim 10^{-8}$ Ω m) is negligible when compared with amorphous carbon (~ 0.01 -0.1 Ω m). (Balandin A. A. *et al. Nat. Mater.* 10, 2011, 569; Giorgia F. *et al. Nano Lett.* 2014, 14, 6109; Chen J. H. *et al. Nat. Nanotech.* 2008, 3, 206; Russo C. J. *et al. J. Struct. Biol.* 2016, 193, 33) The conductivity advances of graphene may help to reduce the heating and charging effects of specimens induced by the electron beam. Therefore, the atomic thickness and excellent conductivity of graphene may contribute to the reduction of beam-induced motion.

Response Figure 5. a, Representative SEM image of suspended graphene membranes at liquid-nitrogen temperature; the grid was examined at the 40° tilt to observe the crinkle. b, The protein motions under radiation on the graphene and the amorphous carbon film. Their protein motions manifest by the average displacement of 20S proteasome particles in ice relative to the initial position versus the density of electron dose.

To make the beam-induced motion on the graphene and amorphous carbon film more statistical, we have accordingly revised the Figure 4b and corresponding method section as follows:

“Fig. 4...b The protein motions under radiation on the graphene and the amorphous carbon film. Their protein motions manifest by the average displacement of 20S proteasome particles in ice relative to the initial position versus the density of electron dose. Each point represents the average displacement of thousands of particles on the graphene grids...”

“...To measure the displacement of particles on various supporting membranes, we collected movies of specimens under Titan Krios (300 kV) with constant dose rate (with accumulated dose of ~ 50 e/Å²), and each micrograph was dose-dependently fractionated to 32 frames. Particles were auto-picked and their coordinates in each frame were determined by Relion3.0. These coordinates were then applied for calculating the displacements of 20S proteasome particles on individual supporting membrane versus dose accumulation...”

10. Lines 187 and 188 mention that 20S proteasomes have a preference for the graphene interface over the air-water interface. Since supplementary Figure 19 b indicates that particles exhibit mainly side views, there may also be a preferred-orientation issue. More information should be provided on this question. In particular, please clarify whether the word “uniform” that is used in line 197 was meant to refer to spatial distribution or to angular distribution. Also please keep in mind that preferential orientation may not result in poor resolution of the 3-D map because of the high point-group symmetry of the proteasome. Readers will be still be very much interested in whether the cylinder-shaped particles are bound to graphene in a highly preferred orientation.

Response:

We thank the referee to raise this concern. The word “uniform” in line 197 refers to the spatial distribution. We have revised the description in the manuscript as “...The robust hydrophilic surface of graphene grids facilitated a uniform and high-density **spatial distribution** of 20S proteasomes...”

We also have included the Euler angle distribution map in the Response Figure 6.

Response Figure 6. Euler angle distribution of 20S proteasomes.

11. Some details are given of conditions used to grow large, single-crystal graphene, including the fact that methane was used as the feedstock. Since the authors emphasize in the title of the manuscript that the graphene is ultraclean, does this mean that the novel, polymer- and transfer-free method described here is sufficient to produce ultraclean graphene, and that copper acetate is not necessary? It would seem to be important for the authors to cite some of recent publications in which copper acetate is used as the feedstock, and to comment on the relative merits of the two choices for making ultraclean graphene.

Response:

We thank the referee to raise this concern. The surface contaminations on graphene devices mainly has two origins: (1) the amorphous carbon introduced during the graphene growth; and (2) the polymer residues induced during the transfer process of graphene. (Lin L. *et al. Nat. Commun.* 2019, 10, 1912; Lin L. *et al. Chem. Rev.* 2018, 118, 9281) Our work aimed to avoid the polymer residues from the conventional graphene transfer, while the work using copper acetate as carbon feedstock was designed to reduce the amorphous carbon rooted in the graphene growth (Jia, K. C. *et al. J. Am. Chem. Soc.* 2019, 141, 7670). Our discussion will proceed from these two aspects.

Recently, our group developed several practical methods to synthesize the super-clean graphene on the copper substrates by reducing the amorphous carbon originated from the graphene growth, including the design of the copper acetate. (Lin L. *et al. Nat. Commun.* 2019, 10, 1912; Jia K. C. *et al. J. Am. Chem. Soc.* 2019, 141, 7670; Zhang, J. C. *et al. Angew. Chem. Int. Ed.* 2019, 58, 14446) In these methods, methane or copper acetate was used as carbon feedstock to grow the super-clean graphene, respectively. However, these ultraclean graphene films were supported on the copper substrates. To fabricate suspended graphene membranes, graphene films need to be placed on the porous substrates.

Conventional methods mainly rely on the transfer of graphene onto the perforated substrates, and polymer films have often been coated on the graphene surface as supports during the previous fabrication techniques. As a result, the inevitable polymer residues would seriously contaminate the graphene surface even if ultraclean graphene film was used as raw material (Supplementary Fig. 12). To address this issue, we developed the polymer-free and transfer-free method to maintain the clean surface of graphene for further application in high-resolution EM. Certainly, a cleaner graphene grid can be achieved when we utilize the super-clean graphene film on the copper substrate as raw material.

To introduce the growth of clean single-crystal graphene to the readers, we have accordingly revised the manuscript as follows: “... **The clean single-crystal graphene films were grown on the commercial copper foil (Alfa-Aesar #46365; a thickness of 25 μm) in the low-pressure CVD system. Firstly, the copper foil was electrochemically polished in the electrolyte solution composed of ethylene glycol and phosphoric acid (v/v = 1:3) with a voltage of 2~3 V for 10-30 min. Then the copper substrate was loaded into the tube furnace and was heated to 1030 with a flow of 100 sccm H_2 in one hour. Subsequently, the copper substrate was annealed at 1030 in 100 sccm H_2 for 30-60 min to remove the surface contamination. Then the H_2 flow was shut down, and the copper foil was heated at a pressure of ~8 Pa for 10~15 min at 1030 . For the growth of graphene, 500 sccm H_2 and 1 sccm CH_4 were introduced and maintained for 3 h. After growth, the copper foil with graphene was pulled out of the tube furnace. After being cooled, the sample was post-treated in 500 sccm CO_2 at ~500 $^\circ\text{C}$ for 3 h to remove the amorphous carbon introduced during the graphene growth, as reported³⁹...**” Accordingly, we have cited the recent publication for growing the super-clean graphene as the referee suggested.

12. Figure 1 f shows data on the “intactness” statistics. Please confirm whether this is for grids to which no sample had yet been loaded. If so, please add information on how

these statistics change after sample is loaded, and, in particular, please add information of how these statistics change when the grids are used to prepare cryo-EM samples.

Response:

Thanks for the reviewer's suggestions. The "intactness" statistics in Figure 1f was for the grids where no sample had been loaded. Thus, we further investigated the statistics change of graphene grids after loading the specimen.

As shown in the Response Figure 7, the average intactness of single-crystal graphene grids decreased from ~90% to ~78%, while the bilayer and few-layer graphene grids maintain their high coverage of graphene membranes (90~100%) after the sample from aqueous solution was loaded. Furthermore, the robust bilayer graphene grids were used to prepare cryo-EM specimens. And the average coverage of graphene membranes remains ~83% for the further data collection (Response Fig. 7e-f).

Response Figure 7. The intactness statistics of graphene grids after sample was loaded. a, Statistical graph showing the intactness of single-crystal graphene grid (~78%), bilayer graphene grid (~90%) and few-layer graphene grid (~95%), respectively. All the grids were treated with the oxygen plasma, then the CuCl_2 aqueous solution was deposited on the graphene grids. b-d, Typical SEM images of single-crystal graphene grid (b), bilayer graphene grid (c) and few-layer graphene grid (d) after sample was loaded. The dark holes in the SEM images revealed the broken graphene membranes. e, Intactness statistics of bilayer graphene membranes after being used to prepare the cryo-

EM samples. The average intactness was ~83% from ~2700 holes. f, Representative low-magnification TEM image of suspended graphene membranes with vitreous ice.

Response to the 2nd Reviewer

Liming Zheng et al present a newly developed and detailed protocol to fabricate ultraclean graphene grids for high-resolution cryo-EM. The importance of graphene grid supports in achieving high resolution structures have been previously shown and sparked a large interest in the cryo-EM community. Despite encouraging initial reports, a reliable and reproducible protocol to yield high-quality grids has been proven extremely challenging.

This study provides evidences that a robust protocol for stable graphene membranes with low background noise and tunable surface properties has been identified. Moreover, the quality and reliability of such grids has been successfully tested by solving the 20S proteasome structure to a remarkable 2.36 Å in resolution.

Together, the study provides a scalable method for graphene grids fabrication that, once available to the cryo-EM community, will facilitate specimen vitrification and improve image quality during cryo-EM data collection.

The manuscript is well written with detailed results that support the conclusion made by the authors.

Response:

We appreciate very much the positive comments from the reviewer on the quality of our work.

Response to the 3rd Reviewer

In my opinion, the manuscript of Zheng et al. "Robust ultraclean atomically thin membranes for atomic-resolution electron microscopy", despite being quite obvious (even I have tried the same approach for preparing ultraclean graphene for TEM grids) is interesting enough for a broad audience and can be eventually published in Nature Communications but only after major revision.

The authors demonstrate the advantages of graphene based membrane grids produced using a polymer- and transfer-free direct-etching method. The resulting graphene membranes are robust, very clean and can be tuned to have different surface hydrophilicity. Demonstrating these advantages in evaluation of actual samples improves the paper and illustrates how negligible background noise and good stability under electron irradiation allows imaging with atomic-resolution including proteins at the graphene-ice interface.

The weaknesses that need to be fixed include poorly written Methods section, especially the details of graphene synthesis, plasma irradiation for hydrophilic modification and others. Some sentences are not even finished, like on line 225. There are also multiple language flaws that could be fixed by a native speaker.

Response:

We appreciate very much the positive comments from the reviewer on the quality of our work and the explicit recommendation of publication. The detailed information of graphene synthesis, plasma irradiation for hydrophilic modification, removal of undesired graphene, photolithography, selective etching, rinsing and drying were added accordingly in the revised Methods section as follows:

Graphene synthesis

The clean single-crystal graphene films were grown on the commercial copper foil (Alfa-Aesar #46365; a thickness of 25 μm) in the low-pressure CVD system. Firstly, the copper foil was electrochemically polished in the electrolyte solution composed of ethylene glycol and phosphoric acid (v/v = 1:3) with a voltage of 2~3 V for 10-30 min. Then the copper substrate was loaded into the tube furnace and was heated to 1030 $^{\circ}\text{C}$ with a flow of 100 sccm H_2 in one hour. Subsequently, the copper substrate was annealed at 1030 $^{\circ}\text{C}$ in 100 sccm H_2 for 30-60 min to remove the surface contamination. Then the H_2 flow was shut down, and the copper foil was heated at a pressure of ~ 8 Pa for 10~15 min at 1030 $^{\circ}\text{C}$. For the growth of graphene, 500 sccm H_2 and 1 sccm CH_4 were introduced and maintained for 3 h. After growth, the copper foil with graphene was pulled out of the tube furnace. After being cooled, the sample was post-treated in 500 sccm CO_2 at ~ 500 $^{\circ}\text{C}$ for

3 h to remove the amorphous carbon introduced during the graphene growth, as reported³⁹.

To synthesize the bilayer graphene, the polished copper foil should be placed on the quartz substrate before loaded into the tube furnace. Then the copper foil was heated to 1030 °C in one hour and annealed for 30 min under a flow of 100 sccm H₂. Subsequently, the 1000 sccm H₂ and 1 sccm CH₄ were introduced into the LPCVD system for ~1.5 h with a pressure of ~2000Pa. Subsequently, 500 sccm H₂ and 1 sccm CH₄ were induced for ~0.5 h. After growth, the graphene can also be treated with CO₂ as mentioned above.

For the synthesis of few-layer graphene, the polished Cu was placed on the quartz substrate and loaded into the LPCVD system. After the heating and annealing process as noted above, the graphene film was grown under the flow of 2000 sccm H₂ and 2 sccm CH₄ for 2 h. After that, the same CO₂ treatment can be used to clean the graphene surface.

Controllable oxygen-plasma treatment

The reactive ion etcher (Pico SLS, Diener) used here can modulate the species of source gas, volume flow rate of gas, plasma treatment time and power. For the controllable oxygen-plasma treatment, the graphene grids were placed directly on a metal substrate in the plasma chamber, then the power was controlled at a low energy of 40W and the volume flow rate was manipulated at a flow of 5 sccm oxygen. By regulating the time of oxygen plasma treatment, desirable wettability and surface modification of graphene grid can be achieved.

Plasma treatment to remove the undesired graphene

To remove the graphene on one side of copper foil, we placed the copper foil with graphene on a smooth hard substrate such as clean glass slide or polyethylene glycol terephthalate (PET) film. Then the scotch tape was used to seal the edges of copper foil on the substrate. In this way, the graphene on the other side of copper foil was protected from being damaged by the following plasma treatment. After placing the hard substrate in the reactive ion etcher (Pico SLS, Diener), a flow of 10 sccm air was introduced, then the air plasma was generated at a power of 150 W for 3 min to remove the undesired graphene.

Photolithography

The positive photoresist (AR-P5350, ALLRESIST) was spun onto the Cu foil at 4000 rpm and baked at 110 °C for 3 min. Subsequently, the positive photoresist was exposed to 365 nm ultraviolet for 6-10 s with the designed mask. After baking at 110 °C for 1 min, the foil was developed in dilute developer (AR 300-26, ALLRESIST; developer: water = 1:7) for 6-10 s. In this way, the patterned photoresist was fabricated on the copper foil.

Selective etching

The exposed area of the patterned copper foil was selectively etched with 0.1M $\text{Na}_2\text{S}_2\text{O}_8$ aqueous solution. Note that a stir bar could be used for controlling shapes of the holes.

Rinsing

The floating graphene grids were washed gently with deionized water to remove etchant traces. And the photoresist was carefully removed with N-Methyl pyrrolidone and acetone, then the grids were rinsed in the isopropyl alcohol.

Drying

The bilayer and few-layer graphene grids were normally dried in the super-clean room to avoid extra contaminants after being took out from the isopropyl alcohol.

The monolayer graphene grids were dried in the critical point drier (samdri-795) to protect the suspended graphene membranes from rupturing by surface tension.

In addition, we thank the referee to point out the mistakes and language flaws in the sentences. We have carefully revised the language of the manuscript carefully. In the following pages, we will address the reviewer's constructive comments.

1. The use of Young-Laplace equation $P_{cap}=2\gamma_{l-g}/R_l$ on page 102 for calculation of the capillary pressure P_{cap} is not supported by illustration of R_l on Figure 1.

Response:

Thanks for the reviewer's valuable suggestions. We have accordingly illustrated the curvature radius of the liquid surface (R_l) in the Figure 1b as follow:

2. It would be helpful to show the survival yield variation with the hole diameter in Figure 1f or Table S1. It is also not clear when normal and when the supercritical drying were used.

Response:

Thanks for the reviewer's constructive suggestions. The survival yield variation with different hole diameters of suspended graphene is significant for the further application of graphene membranes. As shown in the Supplementary Figure 22, the coverage of suspended graphene membranes decreases as the suspended size increases. Owing to the enhanced mechanical strength and reduced stress of bilayer and few-layer graphene membranes, the coverage of suspended graphene shows little degradation and remains ~90% even when the diameter is up to 50 μm . In contrast, the coverage of suspended single-crystal graphene is significantly decreased once the lateral suspended size is over than ~20 μm . Therefore, the smaller suspended size of graphene membrane is recommended to achieve the higher coverage of graphene grid, especially for the monolayer graphene grid.

Furthermore, the supercritical drying can be used to eliminate the liquid surface tension and improve the coverage of suspended graphene membranes. In this work, we used the supercritical drying to fabricate the high-intactness monolayer graphene grids. For the bilayer and few-layer graphene grids, they were robust enough to withstand the surface tension, so they were dried under the normal conditions.

To make the influence of drying method and suspended size more straightforward to readers, we have accordingly revised the manuscript as follows:

“...The bilayer and few-layer graphene grids can reach the intactness as high as 91~100% after being dried in the air...Moreover, the smaller suspended size of graphene membranes also contributes to the lower probability of breakage (Supplementary Fig. 22) ...”

“...Drying: The bilayer and few-layer graphene grids were normally dried in the super-clean room to avoid extra contaminants after being took out from the isopropyl alcohol. The monolayer graphene grids were dried in the critical point drier (samdri-795) to protect the suspended graphene membranes from rupturing by surface tension...”

In addition, we have added one figure (Supplementary Figure 22) in supporting information.

Supplementary Figure 22. Coverage of suspended graphene membranes with different diameters. **a**, Statistical graph showing the coverage of single-crystal, bilayer and few-layer suspended graphene membranes with different suspended sizes. **b**, SEM images of single-crystal suspended graphene membranes with the diameter of $\sim 15\ \mu\text{m}$, $\sim 20\ \mu\text{m}$, $\sim 30\ \mu\text{m}$ and $\sim 40\ \mu\text{m}$, respectively. **c**, SEM images showing the bilayer suspended graphene membranes with the diameter of $\sim 20\ \mu\text{m}$, $\sim 30\ \mu\text{m}$, $\sim 40\ \mu\text{m}$ and $\sim 50\ \mu\text{m}$, respectively. **d**, SEM images of few-layer suspended graphene membranes with the diameter of $\sim 30\ \mu\text{m}$, $\sim 40\ \mu\text{m}$, $\sim 50\ \mu\text{m}$ and $\sim 60\ \mu\text{m}$, respectively.

3. Some data in Figures are not mentioned or poorly described in the text, e.g. reference to Figure 4d is unclear and no definition is given to Gold-standard Fourier-shell correlation (FSC) in Figure 4e.

Response:

Thanks for the reviewer's valuable suggestions. The positions of biomolecules embedded in the vitreous ice on the graphene grid can be detected by the cryo-electron tomography (cryo-ET), including the biomolecules on the air-water and graphene-water interfaces

(Fig. 4d). In this way, the distribution of biomolecules through Z-axis can be revealed from the cryo-ET (Fig. 4c). We found that most 20S proteasomes adsorbed to the graphene-ice interface (Figs. 4c-d), indicating the 20S proteasomes had strong preferences for the hydrophilic graphene surface over the air-water interface.

To make the Figure 4d clearer to the readers, we have accordingly revised the manuscript as follows:

“...Additionally, the positions of 20S proteasomes embedded in the vitreous ice on the graphene grid were detected by the cryo-electron tomography (cryo-ET), including the 20S proteasomes on the air-water and graphene-water interfaces (Fig. 4d). In this way, the distribution of 20S proteasomes through Z-axis can be revealed from the cryo-ET (Fig. 4c). We found that most 20S proteasomes adsorbed to the graphene-ice interface (Figs. 4c-d), indicating the 20S proteasomes had strong preferences for the hydrophilic graphene surface over the air-water interface....”

On the other hand, the Fourier-shell correlation (FSC) is the correlation between two independent maps, where each map is calculated from half the dataset; and FSC=0.143 as a threshold criterion was chose in part to make the resolution in cryo-EM comparable to that measured in X-ray crystallography (Henderson R. *et al. J. Mol. Biol.* 2013, 333, 721). Nowadays, the cutoff FSC=0.143 is widely used as resolution criterion for cryo-EM reconstructions.

To make the Figure 4e clearer to the readers, we have accordingly revised the manuscript as follows:

“...e Gold-standard Fourier-shell correlation (FSC) curve for the reconstructed 20S proteasome supported on the graphene grids, where FSC=0.143 is commonly used as a criterion to calculate the nominal resolution of cryo-EM reconstruction...”

Reviewers' comments:

Reviewer #1 (Remarks to the Author):

The revisions are all excellent, and they add great value to the manuscript.

I nevertheless still have two suggestions for further revision before publication.

1. Unless I am wrong, the grid-atlas image that is now included contains many areas where the ice thickness is much greater than the values shown in the histogram, Supplementary Figure 21. I wonder whether the areas where the reported thickness values were made have been pre-selected to just include the thinnest areas that can be seen in the grid atlas. Doing so makes a lot of sense, as it is what a user would normally do anyway. However, if that is what was done, then a sentence should be added to say so.

2. Unless I have missed it - which is certainly possible, I do not know what the error bars mean in Figure 4, panel b. If is not already explained, please say whether the bars are the standard deviation of the thousands of measurements that go into one point (i.e. the average, shown as a dot), or do the error bars represent the range of values observed over different areas of the grid, or even over multiple grids.

Please also add error bars for the graphene measurements, or at least explain why they are not shown for graphene.

Reviewer #2 (Remarks to the Author):

I support the publication of the revised manuscript, and hope to see these new graphene grids soon available to all EM community.

Reviewer #3 (Remarks to the Author):

The revised manuscript of Zheng et al. "Robust ultraclean atomically thin membranes for atomic-resolution electron microscopy" has been improved but still has numerous confusing statements and descriptions preventing from recommending it for publication in the current shape.

The language flaws, including those in the newly written corrections, could be fixed by a native speaker but the main weaknesses to me remain in the poorly written Methods section -- the details of graphene synthesis, plasma irradiation for hydrophilic modification.

For example:

1. The three types of graphene synthesis, "single crystal", double layer and few layer are described with confusing set of details. Although the apparent difference is in the amount of hydrogen and methane, different additional subtleties are included for different versions to make the readers confused. It is stated only the bilayer graphene synthesis that "the polished copper foil should be placed on the quartz substrate". Was it not in other cases?

2. It was stated only for the bilayer that the synthesis took place at "a pressure of ~ 2000 Pa". Was it not in other cases?

3. It was stated for the bilayer and multilayer synthesis that the graphene "can also be treated with CO₂ as mentioned above." Was it typically clean enough that the treatment was not necessary?

4. It was stated only for "single crystal" synthesis that "the H₂ flow was shut down, and the copper foil was heated at a pressure of ~ 8 Pa for 10~15 min at 1030-". Was it not in other cases?

5. Speaking of the single crystal, it is not obvious to an unprepared reader what causes it to grow such. It is my understanding that epitaxial growth on (111) Cu surface is responsible for that but it is not clear how such foil orientations were achieved. During annealing, domains of different Cu orientations grow in size. The thermodynamically stable (111) orientation often can achieve quite

large sizes but also other orientations, like (100), can be also observed, as these authors reported as well. The authors should be more clear on that. Besides, the single crystal presumes no weak boundaries and, as was shown in Nature Materials 17 (4), 318, the real single crystals of significantly larger sizes survive copper removal without polymer, while the survival rate of 40 mkm in here is only 40%

Response to the 1st Reviewer

The revisions are all excellent, and they add great value to the manuscript.

I nevertheless still have two suggestions for further revision before publication.

Response:

We appreciate very much the positive comments and kind suggestions for our work. To strengthen the presentation of our paper, we will fully address the reviewer's comments and concerns point by point in the following.

1. Unless I am wrong, the grid-atlas image that is now included contains many areas where the ice thickness is much greater than the values shown in the histogram, Supplementary Figure 21. I wonder whether the areas where the reported thickness values were made have been pre-selected to just include the thinnest areas that can be seen in the grid atlas. Doing so makes a lot of sense, as it is what a user would normally do anyway. However, if that is what was done, then a sentence should be added to say so.

Response:

We thank the reviewer for pointing this out. Indeed, the ice of some areas in the grid-atlas is pretty thick, and sometimes reaches ~75 nm as shown in Supplementary Figure 21 (especially those "dark" regions). However, in practice, cryo-EM users would normally avoid collecting datasets in such "dark" regions, and the ice thickness of the majority of the grid areas was suitably thin for data collection.

As suggested by the reviewer, we have added a sentence in Supplementary Figure 21 as follows:

"...The ice thickness measurement was mainly performed at those data collection area that accounted for the majority regions of the grid..."

2. Unless I have missed it - which is certainly possible, I do not know what the error bars mean in Figure 4, panel b. If is not already explained, please say whether the bars are the standard deviation of the thousands of measurements that go into one point (i.e. the average, shown as a dot), or do the error bars represent the range of values observed over different areas of the grid, or even over multiple grids.

Please also add error bars for the graphene measurements, or at least explain why they are not shown for graphene.

Response:

We thank the reviewer for pointing this out. The error bars represent the range of values

observed over different areas of the grid. And the error bars for the graphene measurements are added.

As suggested by the reviewer, we have revised the Figure 4b as follows:

"...and the error bars represent the range of values observed over different areas of the grid..."

Response to the 2nd Reviewer

I support the publication of the revised manuscript, and hope to see these new graphene grids soon available to all EM community.

Response:

We appreciate a lot for the reviewer's explicit recommendation of publication.

Response to the 3rd Reviewer

The revised manuscript of Zheng et al. "Robust ultraclean atomically thin membranes for atomic-resolution electron microscopy" has been improved but still has numerous confusing statements and descriptions preventing from recommending it for publication in the current shape.

The language flaws, including those in the newly written corrections, could be fixed by a native speaker but the main weaknesses to me remain in the poorly written Methods section -- the details of graphene synthesis, plasma irradiation for hydrophilic modification.

Response:

We thank the referee to point out the language flaws and confusing statements in the sentences. We have carefully revised the language of the manuscript. The detailed information of graphene synthesis, plasma irradiation for hydrophilic modification were revised in the Methods section as follows:

Controllable oxygen-plasma treatment

A reactive ion etcher (Pico SLS, Diener) was used to modulate the species of source gas, volume flow rate of gas, treatment time and power (Supplementary Fig. 23). For controllable oxygen-plasma treatment, graphene grids were placed directly on a metal substrate in the plasma chamber. Then the oxygen plasma was generated with a low-energy power of 40 W and a flow of 5 sccm oxygen. By controlling the plasma treatment time, desirable wettability and defect density of graphene grid can be achieved (Figure 3a-b).

Supplementary Figure 23. Reactive ion etcher. The chamber of reactive ion etcher, the

control systems of source gas, volume flow rate, time and power are marked in the optical image, respectively.

Graphene synthesis

The clean single-crystal graphene films were grown on commercial copper foils (Alfa-Aesar #46365; a thickness of 25 μm) in a low-pressure CVD (LPCVD) tube furnace system. Firstly, the copper foil was electrochemically polished in an electrolyte solution composed of ethylene glycol and phosphoric acid (v/v = 1:3) with a voltage of 2~3 V for 10-30 min. The copper substrate was loaded into the tube furnace and heated to 1030 $^{\circ}\text{C}$ under a flow of 100 sccm H_2 in one hour. The annealing of the copper substrate was carried out at 1030 $^{\circ}\text{C}$ in 100 sccm H_2 for 30-60 min to eliminate the surface oxide and contamination. Then the copper foil was annealed at a pressure of ~ 8 Pa for another 10~15 min at 1030 $^{\circ}\text{C}$ when the H_2 flow was shut down. This annealing treatment passivated the active sites of copper surface, which helped to suppress the nucleation density of graphene seeds and facilitated the growth of large-domain single crystalline graphene films. For the growth of graphene, 500 sccm H_2 and 1 sccm CH_4 were introduced into the LPCVD system and maintained for 3 h. After growth, the copper foil covered with graphene films was rapidly cooled down to the desired temperature (450–550 $^{\circ}\text{C}$, 5–10 min) while still under the same H_2 and CH_4 flow. Then, the supply of H_2 and CH_4 were both stopped. For the superclean graphene preparation, CO_2 gas (500 sccm, 3 h) was pumped into the LPCVD system to selectively remove the coating of amorphous carbon from graphene surface at ~ 500 $^{\circ}\text{C}$, as previously reported by our group³⁹.

For the growth of bilayer graphene films, electrochemically polished copper foil (Alfa-Aesar #46365) was placed on a quartz substrate before loaded into the tube furnace. The copper foil was heated to 1030 $^{\circ}\text{C}$ in one hour and annealed for 30 min under a flow of 100 sccm H_2 . Subsequently, 1000 sccm H_2 and 1 sccm CH_4 were introduced into the LPCVD system for ~ 1.5 h with a pressure of ~ 2000 Pa. After growth, the bilayer graphene films were treated with CO_2 etching at 500 $^{\circ}\text{C}$ as mentioned above.

For the growth of few-layer graphene films, electrochemically polished Cu foil was placed on the quartz substrate and loaded into the LPCVD system. Under a flow of 100 sccm H_2 , the copper foil was firstly heated to 1030 $^{\circ}\text{C}$ in one hour and then annealed for 30 min. Subsequently, the graphene film was grown on the copper foil under the flow of

2000 sccm H₂ and 2 sccm CH₄ for 2 h. After that, the same CO₂ treatment was used to clean the few-layer graphene surface.

To strengthen the presentation of our paper, we will fully address the reviewer's comments in the following pages.

1. The three types of graphene synthesis, "single crystal", double layer and few layer are described with confusing set of details. Although the apparent difference is in the amount of hydrogen and methane, different additional subtleties are included for different versions to make the readers confused. It is stated only the bilayer graphene synthesis that "the polished copper foil should be placed on the quartz substrate". Was it not in other cases?

Response:

We thank the referee to raise this concern. Extensive studies have concentrated on the growth of high-quality graphene film over the years. And the growth process can be tuned by many parameters such as the temperature, pressure, amount of carbon precursor, gas composition (both reducing and oxidizing gas), substrate types, and surface structural characteristics (Lin L. *et al. Chem. Rev.* 2018, 118, 9281). Thus, different additional subtleties are included for the synthesis of single-crystal, bilayer and few-layer graphene.

For the synthesis of bilayer and few-layer graphene film, the polished copper foils were placed on the quartz substrates as mentioned in the Methods section, while it was not needed in the synthesis of single-crystal graphene.

2. It was stated only for the bilayer that the synthesis took place at "a pressure of ~2000Pa". Was it not in other cases?

Response:

We thank the referee to raise this concern. The pressure of ~2000 Pa was only used in the growth of bilayer graphene on the copper foil, and it was not necessary in other cases of this work.

3. It was stated for the bilayer and multilayer synthesis that the graphene "can also be treated with CO₂ as mentioned above." Was it typically clean enough that the treatment was not necessary?

Response:

We thank the referee to raise this concern. To synthesize the clean bilayer and multilayer

graphene, the CO₂ treatments were needed to remove the amorphous carbon introduced during the graphene growth.

To make the growth of clean bilayer and multilayer graphene clearer to the readers, we have accordingly revised the manuscript as follows:

“...After growth, the bilayer graphene films were treated with CO₂ etching at 500 °C as mentioned above....”

“...After that, the same CO₂ treatment was used to clean the few-layer graphene surface...”

4. It was stated only for “single crystal” synthesis that “the H₂ flow was shut down, and the copper foil was heated at a pressure of ~8 Pa for 10~15 min at 1030~”. Was it not in other cases?

Response:

We thank the referee to raise this concern. The H₂ flow was shut down to obtain a non-reducing annealing environment, which help to suppress the nucleation density of graphene and facilitate the growth of large graphene single crystals (Wang H. et al. *Adv. Mater.* 2016, 28, 8968; Duan X. F. et al. *Nat. Commun.* 4, 2096; Zhang J. C. et al. *Adv. Mater.* 2019, 1903266). And this treatment was only used in the synthesis of single-crystal graphene in our work.

To make the annealing process of three types of graphene synthesis clearer to the readers, we have accordingly revised the manuscript as follows:

“...The annealing of the copper substrate was carried out at 1030 °C in 100 sccm H₂ for 30-60 min to eliminate the surface oxide and contamination. Then the copper foil was annealed at a pressure of ~8 Pa for another 10~15 min at 1030 °C when the H₂ flow was shut down...”

“...For the growth of few-layer graphene films, electrochemically polished Cu foil was placed on the quartz substrate and loaded into the LPCVD system. Under a flow of 100 sccm H₂, the copper foil was firstly heated to 1030 °C in one hour and then annealed for 30 min. Subsequently, the graphene film was grown on the copper foil under the flow of 2000 sccm H₂ and 2 sccm CH₄ for 2 h. After that, the same CO₂ treatment was used to clean the few-layer graphene surface...”

5. Speaking of the single crystal, it is not obvious to an unprepared reader what causes it

to grow such. It is my understanding that epitaxial growth on (111) Cu surface is responsible for that but it is not clear how such foil orientations were achieved. During annealing, domains of different Cu orientations grow in size. The thermodynamically stable (111) orientation often can achieve quite large sizes but also other orientations, like (100), can be also observed, as these authors reported as well. The authors should be more clear on that. Besides, the single crystal presumes no weak boundaries and, as was shown in *Nature Materials* 17 (4), 318, the real single crystals of significantly larger sizes survive copper removal without polymer, while the survival rate of 40 μm in here is only 40%.

Response:

Thanks for the reviewer's valuable suggestions. Numerous studies from our research group and others have focused on the synthesis of single-crystal graphene over the past ten years. These efforts can be classified into two approaches depending on whether the single-crystal graphene films were grown from one seed or multiple seeds (Response Fig. 1; see details in our review papers recently published on *Advanced Materials* and *Chemical Reviews*: Zhang J. C. et al. *Adv. Mater.* 2019, 1903266; Lin L. et al. *Chem. Rev.* 2018, 118, 9281).

In the single-seed approach, the control of nucleation density is vital for enlarging the domain size of single-crystal graphene. A common strategy to reduce nucleation density is the passivation of active sites on the metal substrate. The introduction of trace amount of oxygen enabled the efficient passivation of active sites on copper foil, contributing to the formation of subcentimeter-sized single-crystal graphene domains (Hao Y. et al. *Science* 2013, 342, 720; Wang H. et al. *Adv. Mater.* 2016, 28, 8968; Duan X. F. et al. *Nat. Commun.* 4, 2096). In our work, the pretreatment of copper foil in non-reducing annealing environment was utilized to reduce the nucleation density (Wang H. et al. *Adv. Mater.* 2016, 28, 8968), and the single-crystal graphene was prepared in the single-seed approach.

For the multi-seed approach, the aligned crystalline orientation and the seamless merging of adjacent domains lead to a large single-crystal graphene film. As the referee mentioned, the Cu(111) substrate was the most common epitaxial substrate for growing the well-aligned graphene domains, because its lattice symmetry matches the lattice of graphene well. To prepare the single-crystal Cu(111) foils, the long-time high-temperature annealing, the temperature-gradient-driven annealing and the contact-free annealing were developed in the recent years (Lee Y. H. et al. *Adv. Mater.* 2015, 27, 1376; Liu K. et al. *Sci. Bull.* 2017, 62, 1074; Ruoff R. S. et al. *Science* 2018, 362, 1021). Yet the controllable annealing for desirable specific Cu orientation is still very challenging.

Besides, the work the referee mentioned is really inspiring for us. The centimeter-size single-crystal graphene without polymer can survive when supported by the liquid surface, and it can be transferred onto the SiO₂/Si wafer (Vlassiuk I. V. *et al.* *Nat. Mater.* 2018, 17, 318). This work further supports our observations in Supplementary Figure 6 and Supplementary Figure 3, where the graphene membranes without polymer support kept intact on the liquid surface. Yet the main challenge of obtaining large-area free-standing graphene lied in the drying process, in which the graphene membranes deformed and the wrinkles cracked due to the liquid surface tension.

To make the growth of single-crystal graphene clearer to the readers, we have accordingly revised the manuscript as follows:

“...Then the copper foil was annealed at a pressure of ~8 Pa for another 10~15 min at 1030 °C when the H₂ flow was shut down. This annealing treatment passivated the active sites of copper surface, which helped to suppress the nucleation density of graphene seeds and facilitated the growth of large-domain single crystalline graphene films...”

Response Figure 1. Single-seed and multi-seed growth of single-crystal graphene. The upper is the single-seed approach, and the suppressed nucleation density is critical for the growth of large-size single-crystal graphene. The lower is the multi-seed approach, where the aligned crystalline orientation and the seamless merging of adjacent domains contribute to a large single-crystal graphene film. Copyright 2019, Wiley-VCH.

Response to the 3rd Reviewer

The revised manuscript of Zheng et al. "Robust ultraclean atomically thin membranes for atomic-resolution electron microscopy" has been improved but still has numerous confusing statements that the authors can probably fix without the need of further reviewing.

Response:

We thank the referee to point out the confusing statements in the Methods section. To strengthen the presentation of our paper, we have further improved the description about graphene growth for the readers in the following pages.

1. I still saw a few language flaws but will focus on the substance. The authors did clarify that the single crystal actually means that there are large size single crystal domains. The size of those domains and their comparison to the sample size would be appropriate.

Response:

Thanks for the reviewer's suggestions. We have carefully revised the language flaws in the Methods section of the manuscript. In addition, the millimeter-sized graphene single crystals were used in this work, the lateral size of single-crystal graphene can be up to 3 mm, which is comparable with the diameter of a TEM grid (~3 mm).

2. The details of synthesis are still confusing. Why the pressure (~2000 Pa) is specified only for the double layer case? I suggest providing the pressure of synthesis for other conditions as well.

Response:

We thank the reviewer's suggestions. The pressure of graphene growth has been found to influence the growth rate, layer number, and quality of the graphene (Lin L. et al. Chem. Rev. 2018, 118, 9281). A high H₂/CH₄ ratio was commonly used to grow a second graphene layer underneath the existing monolayer graphene (Zhou. H. et al. Nat. Commun. 2013, 4, 2096), yet the growth rate was really slow. The high pressure of ~2000 Pa in the CVD system helps to increase the growth rate and the coverage of bilayer graphene. Moreover, the pressures of synthesis system for other growth conditions were added in the Methods section as the reviewer suggested.

To make the growth of bilayer graphene clearer to the readers, we have accordingly revised the manuscript as follows:

“...Not that the high pressure of ~2000 Pa in the CVD system helps to increase the growth rate and the coverage of bilayer graphene...”

3. I do not want to sound picky in ensuring that the details of synthesis are provide unambiguously but it is important for those who might decide to verify it experimentally.

Response:

We thank the referee to raise this concern. The detailed information of graphene synthesis, graphene grids fabrication, plasma irradiation for hydrophilic modification are provided in the Methods section. Thanks to the reviewer's suggestions, our manuscript is clearer to the readers and helps them prepare the home-made graphene grids for high-resolution imaging.